# Crustal structure of the Volgo-Uralian subcraton revealed by inverse and forward gravity modeling

Igor Ognev[1], Jörg Ebbing[2], Peter Haas[2]

[1]Institute of Geology and Petroleum Technologies, Kazan Federal University, 4/5 Kremlyovskaya street, Kazan 420008, Russia

[2] Department of Geosciences, Kiel University, Otto-Hahn Platz 1, Kiel, D-24118, Germany

*Correspondence to*: Igor Ognev (ognev.igor94@gmail.com)

**Abstract.** Volgo-Uralia is a Neoarchean easternmost part of the East European craton. Recent seismic studies of the Volgo-Uralian region provided new insights into the crustal structure of this area. In this study, we combine satellite gravity and seismic data in a common workflow to perform a complex study of Volgo-Uralian crustal structure which is useful for further basin analysis of the area. In this light, a new crustal model of the Volgo-Uralian subcraton is presented from a step-wise approach: (1) inverse gravity modeling followed by (2) 3D forward gravity modeling.

First, inversion of satellite gravity gradient data was applied to determine the Moho depth for the area. Density contrasts between crust and mantle were varied laterally according to the tectonic units present in the region, and the model is constrained by the available active seismic data.

The Moho discontinuity obtained from the gravity inversion was consequently modified and complemented in order to define a complete 3D crustal model by adding information on the sedimentary cover, upper crust, lower crust, and lithospheric mantle layers in the process of forward gravity modeling where both seismic and gravity constraints were respected. The obtained model shows crustal thickness variations from 32 to more than 55 km in certain areas. The thinnest crust with a thickness below 40 km is found beneath the Precaspian basin, which is covered by a thick sedimentary layer. The thickest crust is located underneath the Ural Mountains as well as in the center of the Volgo-Uralian subcraton. In both areas the crustal thickness exceeds 50 km. At the same time, initial forward gravity modeling has shown a gravity misfit of ca. 95 mGal between the measured Bouguer gravity anomaly and the forward calculated gravity field in the central area of the Volgo-Uralian subcraton. This misfit was interpreted and modeled as a high-density lower crust which possibly represents underplated material.

Our preferred crustal model of the Volgo-Uralian subcraton respects the gravity and seismic constraints and reflects the main geological features of the region with Moho thickening in the cratons and under the Ural Mountains and thinning along the Paleoproterozoic rifts, Precaspian sedimentary basin, and Pre-Urals foredeep.

## 1 Introduction

Crustal thickness and thicknesses of individual layers of the Earth's crust play a determining role in estimating the thermal field due to the relative abundance of the radioactive heat-producing elements in the crust (Beardsmore and Cull, 2001;

Bouman et al., 2015; Hantschel and Kauerauf, 2009). This fact is particularly important in the case of the Volgo-Uralian subcraton as it is located underneath the Volga-Ural oil and gas-bearing province with several giant oil fields, where the maturity of the organic-rich rocks is considered to be tightly related to the temperature distribution in the crust (Khasanov et al., 2016; Khristoforova et al., 2008). Therefore, having the knowledge of the Volgo-Uralian crustal structure is the first major step into further basin analysis of the area.

Volgo-Uralia is a large easternmost segment of the East European craton (EEC). It has been regarded as a separate subcraton along with Sarmatia and Fennoscandia starting from the end of the 20th century (Gorbatschev and Bogdanova, 1993). The Volgo-Uralian part of the EEC is mostly embedded in the East European (Russian) platform, and like the rest of the platform, it does not show any significant topographic variations. It represents a flat area with absolute relief heights ranging from 50 to 250 m for most of the territory. Despite the unremarkable topography of Volgo-Uralia, the same does not hold for its lithospheric structure. Different crustal layers of the subcraton show thickness variations in the order of several tens of km (Artemieva, 2007; Artemieva and Thybo, 2013; Mints et al., 2015).

Several recent crustal models which encompass Volgo-Uralia are based for the most part on regional seismic investigations (Artemieva and Thybo, 2013; Mints et al., 2015). Nevertheless, the gravitational field can also be an essential constraint for the Moho depth especially on the areas devoid of seismic data or with moderate seismic coverage (e.g. Aitken et al., 2013; Steffen et al., 2017). Nowadays, due to the advent of satellite gravimetry, it is possible to obtain gravity field maps with uniform coverage for almost any desired territory of the Earth with a resolution sufficient for regional Moho depth investigation (Bouman et al., 2015).

In this paper, we present a novel model of the Volgo-Uralian subcraton's crustal structure based on inverse and forward 3D gravity modeling with seismic constraints. The main objective of the study is to build a regional crustal model of Volgo-Uralia which in turn can serve as a basis for the further geothermal modeling of the area. In this paper, Section 2 is devoted to a brief overview of the tectonic setting and history of the region. Section 3 gives an outlook on the methods and datasets that were used in the study. All the used datasets are outlined in Section 3.1. Applied gravity inversion methods are discussed in Section 3.2, which is followed by Section 3.3 where the process of forward gravity modeling is described. The main results of the inverse and forward gravity modeling as well as the final crustal model of Volgo-Uralia and its comparison to other existing models are presented and discussed in Section 4.

## 2 Tectonic setting of the Volgo-Uralian subcraton

The preset-day tectonic setting of the Volgo-Uralian region has formed throughout the assembly of the EEC. It started with the collision of Volgo-Uralia and Sarmatia at 2.1-2.05 Ga which led to the creation of a megacontinent Volgo-Sarmatia with Volga-Don collisional orogen developed on the junction zone between the two segments (Bogdanova et al., 2016). Later, during Meso- and Neoproterozoic times, the Pachelma aulacogen was formed along the Volgo-Uralia-Sarmatia border which in combination with the Precaspian sedimentary basin now delineates the south-western border of the Volgo-Uralian subcraton

(Fig. 1). After several hundred million years, at 1.8 Ga, the collision between Volgo-Sarmatia and Fennoscandia commenced. It ended during the formation of the Rodinia supercontinent at 1.0 Ga. The suture intervening Fennoscandia and Volgo-Sarmatia was the place of Central-Russian orogeny growth which then was reworked into Central-Russian and Volyn-Orcha rifts. At present, the Central-Russia rift system represents the north-western border of the Volgo-Uralian subcraton (Bogdanova et al., 2016). On the east, Volgo-Uralia is separated from the West Siberian basin by the young Late Paleozoic Uralide orogen and Late Proterozoic Timanide orogen (Artemieva, 2007).

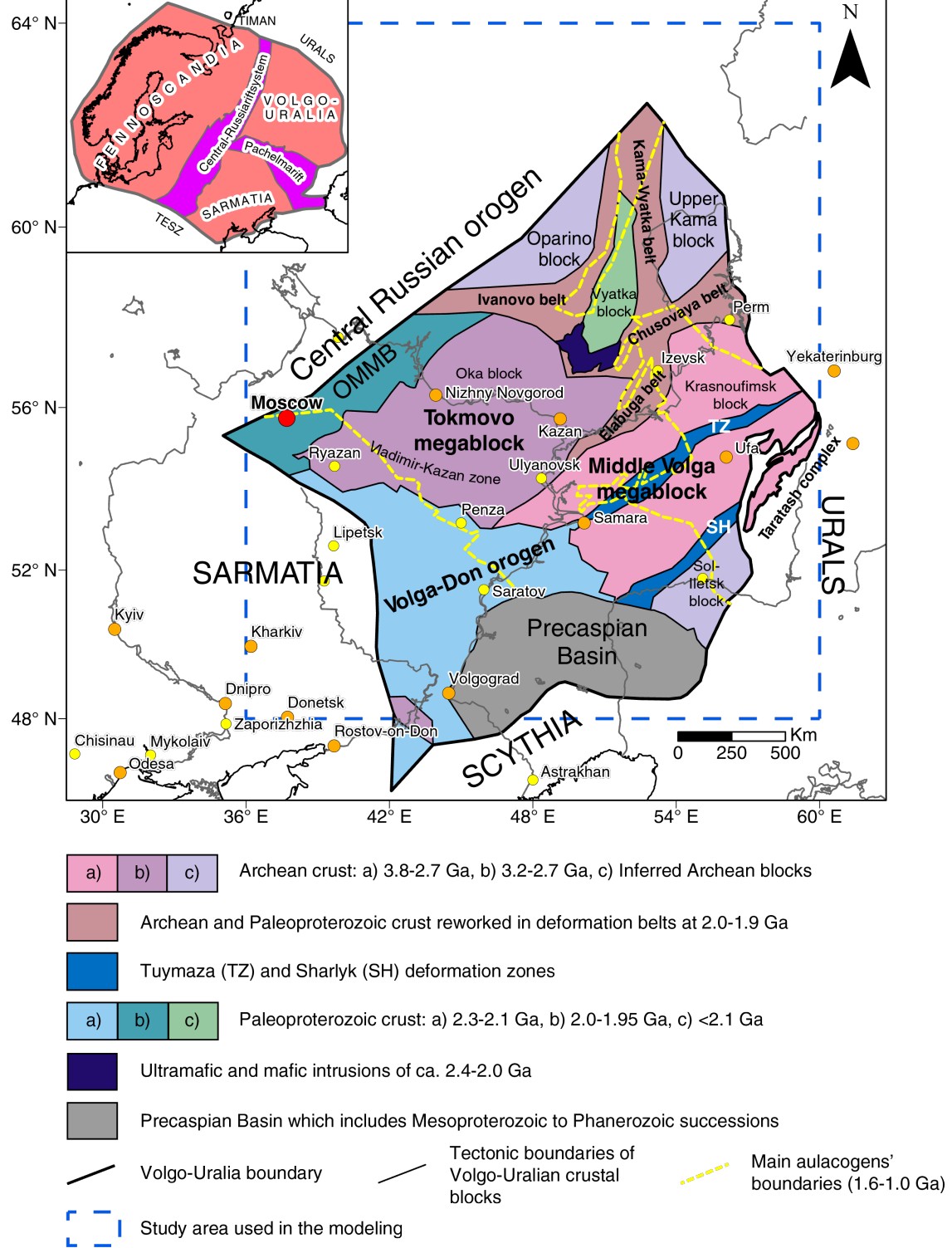

**Figure 1: Main tectonic elements of Volgo-Uralian subcraton (redrawn after Bogdanova et al., 2016).**

In contrast to Sarmatian and Fennoscandian segments of the EEC, Volgo-Uralia except for the Taratash complex is completely covered by Neoproterozoic to Phanerozoic sediments which prevent direct studies of the rocks from the outcrops. Nonetheless,

extensive drilling activity due to the high hydrocarbon potential of the region has provided numerous core samples of the basement which are telling the composition and the age of the Volgo-Uralian segment (e.g. Bogdanova et al., 2010).

For the most part, Volgo-Uralia is comprised of Archean continental crust, which is concentrated in large blocks surrounded by Paleoproterozoic mobile belts. The two most prominent blocks of Archean crust are the Meso- to Neoarchean Tokmovo megablock and Paleo- to Neoarchean Middle-Volga megablock which in the literature are often associated with the so-called

"ovoid" patterns of geophysical anomalies (Bogdanova et al., 2016; Mints et al., 2010). These blocks are dismembered by Elabuga and Chusovaya deformation belts and correspond to relative crystalline basement highs. The sedimentary thickness of the Archean part of Volgo-Uralia rarely exceeds 2 km. Local increases in thicknesses of sedimentary cover are observed in Paleoproterozoic aulacogenic and graben-like structures and can reach up to 5-10 km (Shargorodskiy et al., 2004). A regional trend of a considerable increase of sedimentary cover thickness is observed towards the Ural Mountains in the system of Kama-

Belsk rifts (Fig. S1 in the supplement). Especially thick sedimentary sequences are located to the south of the Volgo-Uralian subcraton where it reaches the Precaspian depression. There sediments have accumulated in successions with a thickness of more than 20 km (Artemieva and Thybo, 2013).

Stratigraphically, the oldest sediments that have accumulated on the Volgo-Uralian territory are of the late Proterozoic age. They can be found sporadically in deep aulacogenic structures within the cratonic area or the deepest zones of large depressions

like the Precaspian basin (Postnikov, 2002; Muslimov et al., 2007). The most extensive sedimentation started in the Middle Devonian and was present throughout the Carboniferous and Early Permian periods. Mesozoic sequences were developed in the north-western and southern peripheries of Volgo-Uralia. Cenozoic sediments are present only in the southern part of the region (Postnikov, 2002). Active Paleozoic sedimentation in concert with subsequent vertical tectonic movements during the Alpine orogenic phase led to the formation of large arch-like structures surrounded by various troughs. These structures now

shape the geometry of sedimentary successions on the Volgo-Uralian subcraton (Bogdanova, 1986; Muslimov et al., 2007). Ones of the most prominent structures are the North and South Tatar arches (Fig. S2). These are the gently sloping uplifts of the sedimentary sequences which have started to form in the Middle-to-Late Devonian (Bogdanova, 1986). During the Paleozoic time, they were mostly the place for marine carbonate sedimentation which was rarely interrupted by upward tectonic movements with marine regression and terrigenous rocks formation. Both North and South Tatar arches reside on the relatively

uplifted crystalline basement and are reflected in the topography (Burov et al., 2003). Moreover, the South Tatar arch despite being similar to its northern counterpart is a very peculiar structure itself. Not only it is an outstanding petroleum-bearing region that holds the giant Romashkino oil field, but it is also a place of active fluid circulation in-between the sediments and the crystalline crust as indicated by Plotnikova (2008). The fact of fluid circulation is supported by several phenomena one of which is decreased density of oil within the South Tatar arch that could result from outgassing of the crystalline basement

(Plotnikova, 2008). Overall, it can be said that sedimentary structures of Volgo-Uralia are linked to the fault block structure of the crystalline basement which is partly inherited from the old Precambrian crustal complexes (Postnikov, 2002).

In terms of the crustal structure, Volgo-Uralia is generally a realm of thick and dense crust principally in its Archean part (Bogdanova et al., 2016). Locally crustal thickness can reach depths up to 60 km in the center of the craton. The evidence of such thick crust in Volgo-Uralia is found in the recent seismic survey of Tatarstan where several crustal roots plunging to depths of more than 55 km are disclosed on the TATSEIS-2003 reflection profile (Artemieva and Thybo, 2013; Trofimov, 2006). Relatively shallow Moho was observed seismically within the Central Russian and Pachelma Paleoproterozoic rifts representing suture zones between individual segments of the EEC. Another region with documented thin crust is the Precaspian sedimentary basin where the crust is thinning down to 32–36 km (Artemieva, 2007). The recent seismic model EUNAseis suggests that Volgo-Uralia has a thick upper crust (with thickness of more than 30 km in some places) which is associated with the above mentioned crustal roots (Artemieva and Thybo, 2013). Earlier findings reveal the correlation between the thicknesses of crustal layers and the tectonic history of the region. That is to say, there is the thickening of the upper crust along the Central-Russia Paleoproterozoic rift system and the thickening of the lower crust beneath the Archean blocks of the subcraton (Artemieva, 2007).

## 3 Data and methods

The work was subdivided into two main steps to build a crustal model of Volgo-Uralia:

1.      Gravity field inversion where a preliminary estimate of the Moho depth boundary is obtained (see a detailed description in Section 3.2).

2.      3D forward gravity modeling where an extensive crustal model of Volgo-Uralia is built. The model incorporates sedimentary, crustal, lithospheric mantle, and asthenospheric layers along with the previously obtained Moho interface (see a detailed description in Section 3.3).

Before the inversion, the gravity data was preprocessed by calculating and subtracting the sedimentary cover effect from the topographically corrected vertical gravity gradient anomaly. The schematic workflow of the study is shown in Fig. 2.

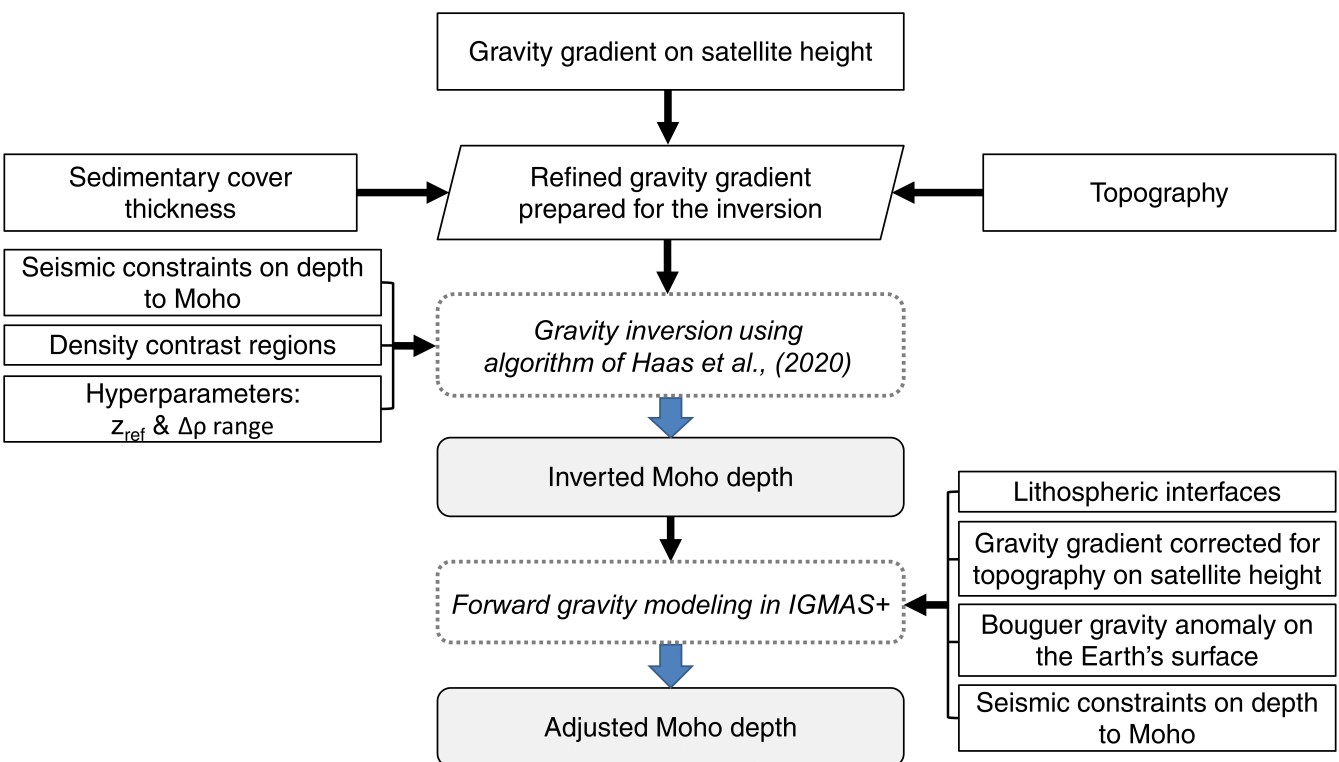

Figure 2: Schematic workflow of the study. The initial step is to prepare the gravity data for the inversion by subtracting the sedimentary cover effect from the topographically corrected vertical gravity gradient. Then it can be followed by a subsequent gravity inversion with laterally variable crust-mantle density contrast (Haas et al, 2020). The inverted Moho depth is incorporated along with other lithospheric interfaces as well as seismic and gravity constraints in IGMAS+ software (Schmidt et al., 2020; Götze and Lahmeyer, 1988). The interfaces are modified to fit the existing gravity and seismic constraints providing the adjusted Moho depth of the region.

## 3.1 Data description

For a successful crustal model construction four main groups of data were utilized:

- Seismic data used to constrain the Moho during the inverse and forward gravity modeling;

- Gravity data used as a main source of information for gravity inversion and one of the constraints in the forward modeling;

- Structural data, used for inverse and forward gravity modeling;

- Petrophysical data, which were implemented in the forward gravity modeling process.

A summary of the used datasets with their sources is given in Table 1.

**Table 1. Summary of datasets used in the modeling**

| Data | Reference |
|---|---|
| **Seismic data** | |
| USGS global seismic catalog | Chulick et al. (2013) |
| TATSEIS-2003 reflection profile | Trofimov (2006) |
| URSEIS-95 reflection profile | Tryggvason et al. (2001), Puchkov (2010) |
| UWARS reflection profile | Thouvenot et al. (1995) |
| ESRU reflection profile | Brown et al. (2002), Rybalka et al. (2006) |
| **Gravity data** | |
| GOCE vertical gravity gradient | Bouman et al. (2016) |
| XGM2019e gravity field model | Zingerle et al. (2019) |
| **Structural data** | |
| ETOPO1 relief | Amante and Eakins (2009) |
| EUNAseis sedimentary thickness | Artemieva and Thybo (2013) |
| LAB interface | Artemieva (2019) |
| **Petrophysical data** | |
| Constraints on sedimentary, crustal, and mantle densities | Artemieva (2007) |

### 3.1.1 Seismic data

Seismic estimations of crustal thickness play a crucial role in gravity modeling as they are the main constraint on the crustal structure. We used seismic data within the studied region from the USGS global seismic catalog (Chulick et al., 2013) which has the information on crustal thickness from the main reflection and refraction surveys performed on the Russian platform mostly during the Soviet period. We also added data coming from recent regional seismic surveys made at the end of the 20[th] and beginning of the 21[st] century on the Volgo-Uralian subcraton which were not originally included in the catalog. These are TATSEIS-2003 geotraverse (Trofimov, 2006) going through the center of Volgo-Uralia, and URSEIS-95, ESRU, and UWARS profiles which mark the crustal structure on the eastern border of Volgo-Uralia crossing the Ural Mountains (Brown et al., 2002; Thouvenot et al., 1995; Tryggvason et al., 2001). Moho depth estimations from seismic databases used in the study are shown in Fig. 3.

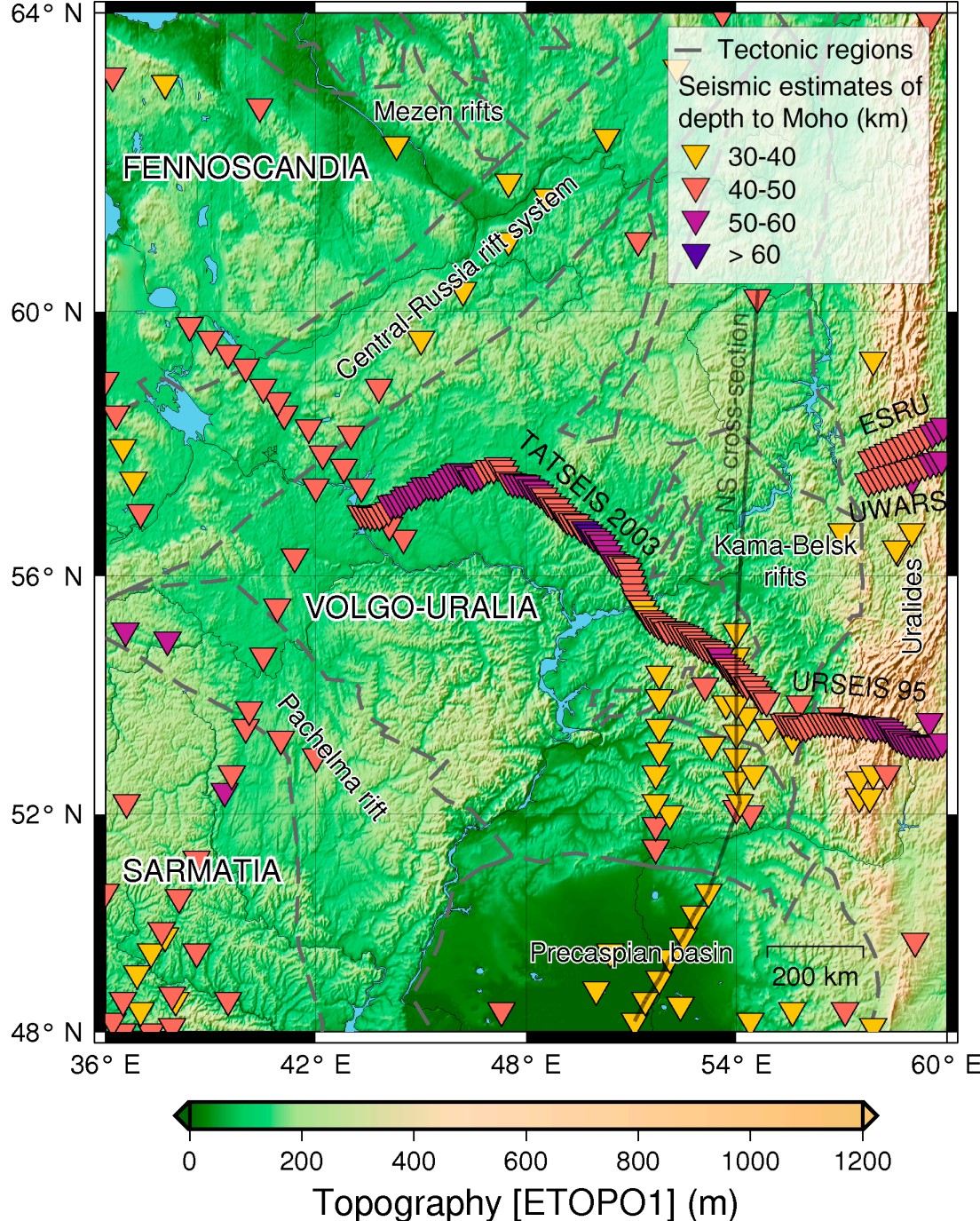

Figure 3: Framework of the studied region with the seismic constraints on Moho depth. Relief is taken from the ETOPO1 model (Amante and Eakins, 2009). Seismic estimates of depth to Moho are used according to USGS seismic catalog (Chulick et al., 2013), TATSEIS-2003 (Trofimov, 2006), URSEIS-95 (Puchkov, 2010; Tryggvason et al., 2001), ESRU (Brown et al., 2002; Rybalka et al., 2006), and UWARS profiles (Thouvenot et al., 1995).

### 3.1.2 Gravity data

In the present workflow, the gravity field is the main source of information used for crustal thickness estimation in the area devoid of seismic constraints. It was shown that GOCE gravity gradients on satellite height are sensitive to interfaces with large density contrasts like Moho (Bouman et al., 2015). That is why we utilized topographically corrected GOCE vertical gravity gradient grids on satellite height of 225 km altitude in the process of gravity inversion (Bouman et al., 2016). In addition, the same topographically corrected GOCE vertical gravity gradient was utilized as a constraint for the forward gravity modeling along with the surface simple Bouguer gravity anomaly from the global gravitational model XGM2019e (Zingerle et al., 2019).

### 3.1.3 Structural data

Several complementary structural datasets were used in the modeling. Surface relief and sedimentary cover thickness are necessary to subtract the gravitational effect of sediments from the topographically corrected vertical gravity gradient field and prepare the gravity data for the inversion (Section 3.2). For that purpose, we took ETOPO 1 topographic model (Amante and Eakins, 2009) and sedimentary cover structure inferred from the EUNAseis seismic model for Moho and crustal structure in Europe, Greenland, and the North Atlantic region (Artemieva and Thybo, 2013).

Knowing the structure of the Earth's lithosphere can also be useful in the forward gravity modeling process as lithosphere-asthenosphere boundary (LAB) is an interface with a density contrast that affects the gravity field. Here, we added the LAB boundary calculated from the concept of thermal isostasy by Artemieva (2019). Being an isothermal boundary, it does not only serve just as additional density contrast but also provides information about the thermal state of the lithospheric mantle.

### 3.1.4 Petrophysical data

The main petrophysical parameter which is involved in operations with gravity field is density. The density model used in the study is given in table 2. Densities of sediments were described by the function of exponential growth of density with depth obtained for the EEC (Artemieva, 2007). Densities of the upper and lower crust were taken based on the seismic estimates of the densities from the CRUST 1.0 model (Laske et al., 2013).

**Table 2. Density model used in the study**

| Layer | Density, kg m$^{-3}$ |
| --- | --- |
| Sedimentary cover | $2430 \times z^{0.045}*$ |
| Upper crust | 2750 |
| Lower crust | 2900 |
| Upper mantle | 3230 |
| Asthenosphere | 3225 |

* z – ½ of the sedimentary strata middle depth in km

Upper mantle density was calculated taking into account the contribution of thermal expansion to the density variations in the subcrustal lithosphere assuming that the average lithospheric mantle temperature is a mean temperature between the temperature at the Moho and temperature at the LAB:

$$\rho_{m\,in\,situ} = \rho_m \left(1 - \alpha \frac{T_M + T_0}{2}\right), \tag{1}$$

where $\rho_m$ – density of the lithospheric mantle at standard conditions, kg m$^{-3}$;

$T_M$ – temperature at the Moho boundary, ºC;

$T_0$ – temperature at the LAB, ºC;

$\alpha$ – thermal expansion coefficient, ºC$^{-1}$.

In this study, we consider that the Archean upper mantle is depleted in mafic components which lowers its density (Kaban et

al., 2003). We take the density of the lithospheric mantle of EEC at room conditions of 3340 kg m$^{-3}$ which corresponds to the Paleoproterozoic-Archean age (Artemieva, 2007). The temperature at the Moho here is taken as 500 ºC which is within the temperature range of Archean-Paleoproterozoic crust of EEC according to (Artemieva, 2007), and LAB temperature as 1400 ºC as in our modeling the thermal LAB model of Artemieva (2019) was utilized. It should be noted that normally Moho temperature should depend on crustal thickness (e.g. Mareschal and Jaupart, 2013). Therefore, a constant Moho temperature

is a simplification accepted in this study. The thermal expansion coefficient is taken as 3.5×10$^{-5}$ ºC$^{-1}$ (Artemieva, 2007, 2019). Using these parameters, we obtained in situ density of the lithospheric mantle as 3230 kg m$^{-3}$.

As the temperature at the Moho boundary does not contribute to the thermal expansion of the asthenosphere, we can slightly modify Eq. (1) to get in situ density of the asthenosphere by taking asthenosphere temperature as equal to LAB temperature:

$$\rho_{a\,in\,situ} = \rho_a \left(1 - \alpha T_0\right), \tag{2}$$

where $\rho_a$ – density of the asthenospheric mantle at standard conditions, which was taken as 3390 kg m$^{-3}$ (Artemieva, 2007). Asthenosphere density is equal to 3225 kg m$^{-3}$. This leads to a quite moderate density contrast between the lithospheric mantle and the asthenosphere of 5 kg m$^{-3}$ which will not have a big impact on the results of forward gravity modeling.

### 3.2 Gravity field inversion

### 3.2.1 Gravity data processing

Gravity field inversion requires initial gravity data to be refined to leave only the gravity signal of interest. In our case, the desired crustal interface is the Moho boundary. In order to obtain the signal that is produced primarily by the Moho undulations, several corrections to the gravity field must be applied. These necessarily would include correction for the latitude, free-air correction, and topographic correction. All the listed corrections are taken into account in the topographically corrected gravity

gradient anomaly. We use topographically corrected vertical gravity gradient for the region with 2670 kg m$^{-3}$ rock density and 1030 kg m$^{-3}$ water density.

Another important interface with high-density contrast that causes anomalies on the satellite gravity field of the same wavelength as Moho does is the sediments-upper crust boundary (Steffen et al., 2017). Volgo-Uralia despite not having a large variation in sedimentary thickness in its cratonic part, is neighbored by Pre-Uralian through and Precaspian basin where sedimentary successions can locally reach up to 10-20 km thickness (Artemieva and Thybo, 2013; Neprochnov et al., 1970). Therefore, it is essential to subtract the gravity effect of sediments from the topographically corrected gravity gradient to get the refined gravity gradient signal produced by the Moho interface:

$$G_{REFINED} = G_{TC} - G_{SED} \, , \tag{3}$$

where $G_{REFINED}$ – gravity gradient field prepared for the inversion which reflects mostly the Moho signal, eotvos;

$G_{TC}$ – topographically corrected gravity gradient, eotvos;

$G_{SED}$ – gravity gradient effect of sediments, eotvos.

As the modeled area is considerably large, we utilized tesseroids to account for the sphericity of the Earth (Uieda et al., 2016). First, the depth of the sediments-upper crust interface was calculated on 1×1-degree mesh using the relief from ETOPO1 and sedimentary thickness from the EUNAseis model. Second, the sedimentary cover was subdivided into a number of tesseroids with lateral dimensions of 1×1 degree and vertical thickness of 1 km. Third, each tesseroid was assigned a certain density depending on its depth. The density-depth relationship for sedimentary cover for the East European platform was taken from (Artemieva, 2007):

$$\rho = 2430 \cdot z^{0.045} \, , \tag{4}$$

where $z$ – ½ of the tesseroid middle depth in km.

Lastly, the gravity effect of sediments was calculated using Tesseroids Python package and it was consequently subtracted from the topographically corrected gravity gradient (Fig. 4).

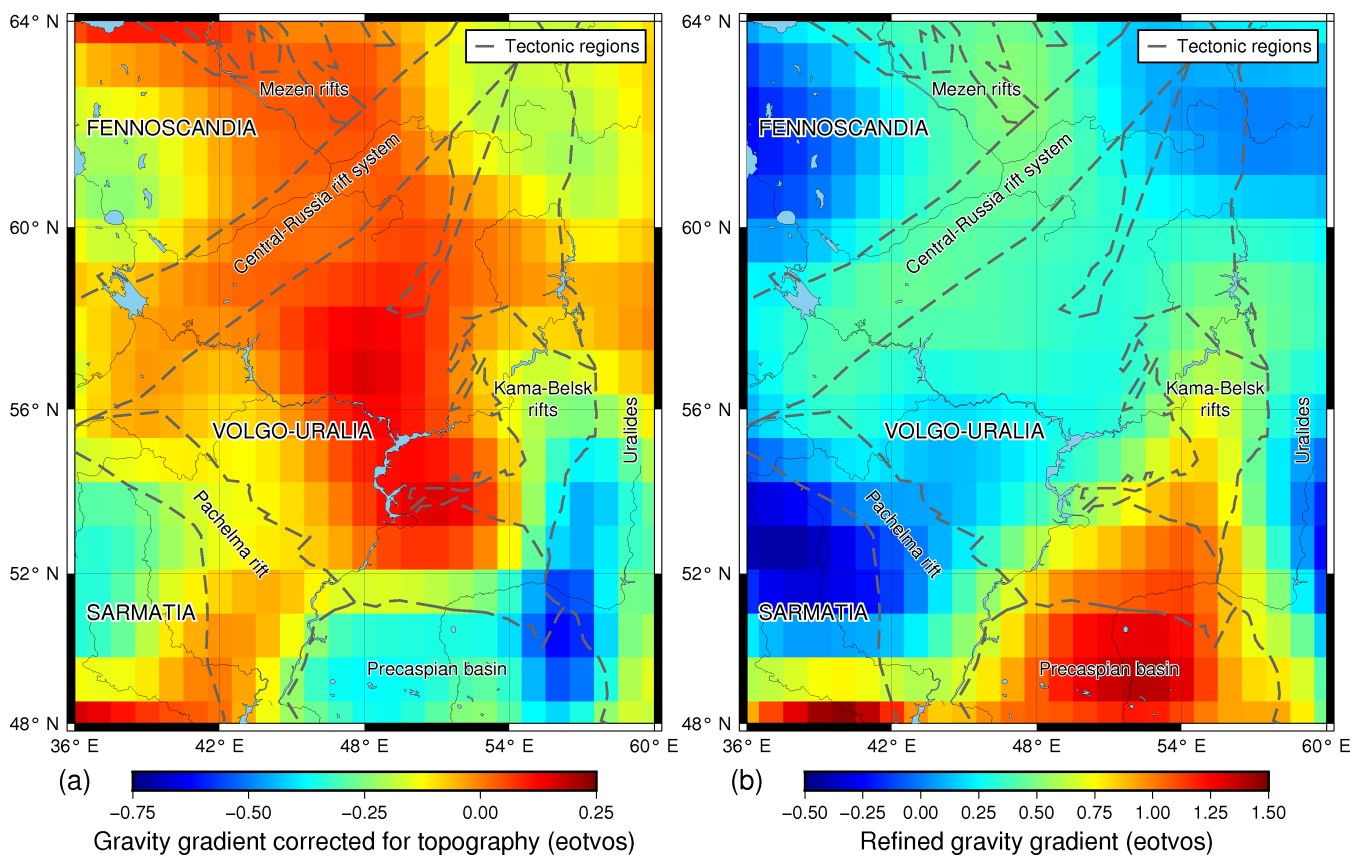

**Figure 4: (a) Vertical gravity gradient corrected for topography on the area of Volgo-Uralian region and (b) refined gravity gradient corrected for both topography and the gravity gradient effect of sediments. Dashed polygons represent tectonic regions used in gravity inversion.**

### 3.2.2 Gravity field inversion with laterally variable density contrast

For the gravity field inversion, we followed a novel approach of Haas et al. (2020) which allows laterally variable crust-mantle density contrasts according to the tectonic regions present in the area of study. This approach solves the inverse problem with the Gauss-Newton algorithm, uses second-order Tikhonov regularization to ensure the stability of the solution, and requires two hyperparameters for the inversion: reference Moho depth $z_{ref}$ and crust-mantle density contrast $\Delta\rho$ with the range of possible values. The algorithm iteratively sets each density contrast from the given range to the predefined tectonic regions keeping spatially constant $\Delta\rho$ within each region. Thus every possible combination of density contrasts' lateral distribution is being checked. Then the combination which gives the smallest RMS error between the Moho depth estimated through the inversion and Moho depth defined at the locations of available seismic measurements is chosen automatically.

Although one can use any gravitational component for the inversion in the abovementioned algorithm, we stuck to the vertical gravity gradient as it is shown to be more sensitive to the Moho undulations than the other components (Bouman et al., 2016).

Here for the purpose of tectonic regionalization, we take the main crustal provinces of Volgo-Uralia from (Bogdanova et al., 2016) which include the Archean cratonic continental crust and Paleoproterozoic mobile belts. We also distinguished Uralide orogen in a separate tectonic region because of its relatively young age and distinct crustal composition. Another tectonic region that deserves our attention is the Precaspian sedimentary basin. The sedimentary strata in its central part reach 20 km of total thickness and include layers of Permian salt with thickness reaching ca. 4-5 km (Volozh et al., 2003; Brunet et al., 1999) which is unique for the EEC. We distinguished this region as a fourth tectonic unit used for gravity inversion.

For the density contrasts, we chose a range of 350 to 550 kg m$^{-3}$ with a step of 50 kg m$^{-3}$. Such a range was chosen according to the previous findings for the region based on satellite gravity (Eshagh et al., 2016) and seismic studies (Rabbel et al., 2013) which suggested that the density for the region should be between 300 and 600 kg m$^{-3}$. We ran the code 10 times using different reference Moho depths ranging from 41 to 50 km with a 1 km step. The range was taken around the average Moho depth of 45 km suggested by the seismic measurements. Before performing the inversion, we decreased the resolution of the used seismic Moho estimates on the newly digitized profiles to make it similar to the resolution of the USGS seismic catalog in the study area. The resolution was decreased 4 times: from 10 to 40 km. Finally, the Moho which fitted best to the seismic constraints was selected. All the calculations during the gravity field inversion were performed on a 1×1-degree grid.

### 3.3 Forward gravity modeling

Gravity inversion was followed by the forward gravity modeling which was done in the IGMAS+ software (Götze and Lahmeyer, 1988; Schmidt et al., 2020). IGMAS+ is a geophysical package aimed at 3D numerical modeling, visualization, and interpretation of potential fields. It offers users to combine different sources of data in a common workflow such as seismic constraints, first, and second-order derivatives of gravitational potential, magnetic field data, and other geological and petrophysical information to produce the most accurate model of the Earth's interior.

At the beginning of the modeling, the study area was laterally extended by 2500 km to minimize edge effects. This has been done by extending the thicknesses of the modeled layers from the edges of the study area. The dimensions of the modeled study area are 2672 km E-W × 3236 km N-S which means that the taken lateral extension is approximately 95 % of the model's latitudinal and 75 % of its longitudinal dimensions. The vertical depth of the model was chosen to be 300 km in order to include all the interfaces along which the main density contrasts arise starting from the bottom of sediments and finishing with the LAB. The 3D model is constructed by triangulated polyhedrons in-between 67 vertical cross-sections, which are oriented in the west-east direction. The approximate distance separating the sections is 50 km. IGMAS+ allows to forward calculate the gravity field from the model and cross-compare it with the measured values which allows to enhance the model in the places of significant gravity misfit. The full process of forward gravity modeling of the current study can be described in 5 steps:

1.       We imported seismic, structural, and gravity data in IGMAS+: (a) Moho interface derived previously from the gravity inversion, (b) depth of the sediments from EUNAseis model (Artemieva and Thybo, 2013), (c) the depth of the LAB interface obtained from the thermal isostasy method (Artemieva, 2019), (d) Available seismic estimates of the Moho depth from USGS seismic catalog, TATSEIS, URSEIS, UWARS, and ESRU seismic profiles (Brown et al., 2002; Chulick et al., 2013;

Thouvenot et al., 1995; Trofimov, 2006; Tryggvason et al., 2001), (e) Bouguer gravity anomaly from XGM2019e global gravity field model (Zingerle et al., 2019), (f) topographically corrected gravity gradient calculated from the gravity gradient grids (Bouman et al., 2016). Additionally, we subdivided the crust into upper and lower parts with the initially horizontal interface. The densities of all the layers were set to the values according to Table 2. The sedimentary layer was discretized in a number of isometric voxels with a 500 m thickness. It allowed representing the exponential increase of sediments' densities with depth.

2.		We adjusted the structure of gravity inverted Moho boundary where seismic data exposed different depths and when it led to the enhancement of the gravity fit or when the seismic data showed consistently different Moho depths on one of the digitized profiles.

3.		We forward calculated gravity and gravity gradient fields from the current model and observed a significant gravity misfit of ca. 95 mGal in the center of the Volgo-Uralian subcraton. This misfit was attributed to the underplated body with a relatively higher density located in the lower crust (see Section 4.2).

4.		We estimated mass imbalance (surplus and deficit) in the area by isostatic calculations following the approach of Ebbing (2007) for the Scandinavian mountain chain:

$$\rho_{sed}D_{sed} + \rho_{UC}D_{UC} + \rho_{LC}D_{LC} + \rho_m D_m + \rho_a D_a - \sum_{i=1}^{5}\rho_{refi}D_{refi} = \Delta Load/g \ , \tag{5}$$

where $\rho$ and $D$ – densities in kg m$^{-3}$ and thicknesses in m of the sedimentary, upper crustal, lower crustal, lithospheric mantle, and asthenospheric layers of the IGMAS+ geological model;

$\rho_{refi}$ and $D_{refi}$ – densities in kg m$^{-3}$ and thicknesses in m of the reference model which are equal to the average values of these parameters used for the corresponding layers in the geological model;

$\Delta Load$ – isostatic load reflecting the mass surpluses and deficits in the area, kg m$^{-1}$s$^{-2}$;

$g$ – normal gravity field, m s$^{-2}$.

The density of the underplated body was set to 3100 kg m$^{-3}$ which gives the difference between the lower crustal density and the assumed body of -200 kg m$^{-3}$. We divided the obtained mass imbalance from Eq. (5) by the difference in densities of the lower crust and the assumed underplated body. In this way, we obtained the thickness of the high-density lower crustal layer associated with the underplating.

5.		The last step was to modify the geometry of the layers to reach a good fit to the gravity data. Here Moho boundary and upper-lower crust interface were subjected to further modifications. The upper-lower crust interface was modified in order to both provide better gravity fit and resemble the patterns of the bottom of the "felsic-intermediate" crust from the EUNAseis model. The Moho was modified in areas of no seismic constraints to enhance the gravity fit.

## 4 Results and discussion

As a result, a new crustal model of the Volgo-Uralian subcraton was obtained throughout the gravity field inversion and forward gravity modeling.

### 4.1 Results of the gravity inversion

In the gravity inversion two hyperparameters, the reference depth and the density contrast, were estimated such that the resulting gravity-inverted Moho showed the minimum RMSE with the seismic Moho depth estimates.

The reference depth which gave the best-fitted Moho to the seismic data was equal to 45 km. Such a relatively deep estimate was obtained due to the fact that TATSEIS-2003 and URSEIS-95 seismic profiles provided a considerable fraction of Moho depths' measurements of more than 50 km. In terms of the density contrast, Archean cratonic crust and Uralide orogen resulted in a density contrast of 550 kg m$^{-3}$, whereas for the Paleoproterozoic belts it was equal to 500 kg m$^{-3}$, and 350 kg m$^{-3}$ for the Precaspian basin (Fig. 5a). These values are close to the previous findings of Eshagh et al. (2016) who used GOCE gravity gradients and determined that crust-mantle density contrast on the territory of Eurasia should be in the range of 400-600 kg m$^{-3}$. At the same time, other seismic-based studies suggest a slightly smaller density contrast around 300-400 kg m$^{-3}$ for the tectonic settings similar to the ones of the modeled region (Chulick et al., 2002; Rabbel et al., 2013). This misfit can arise because gravity-based methods average the crustal and subcrustal densities and express their difference in one signal. Whereas, seismic-based methods restore densities for specific layers in the crust and the lithosphere and give a more targeted look at the contrast in densities between the lower crust and the lithospheric mantle. Our density model used in the forward gravity modeling gives a density contrast of around 330 kg m$^{-3}$ (Table 2) which is closer to the values coming from the seismic-based estimates.

The obtained gravity-inverted Moho depth map generally respects the main known structural features of the crust in the region: Moho thickens in the cratons and Uralides, and thins along the Paleoproterozoic rifts, Pre-Urals foredeep, and Precaspian sedimentary basin (Fig. 5b).

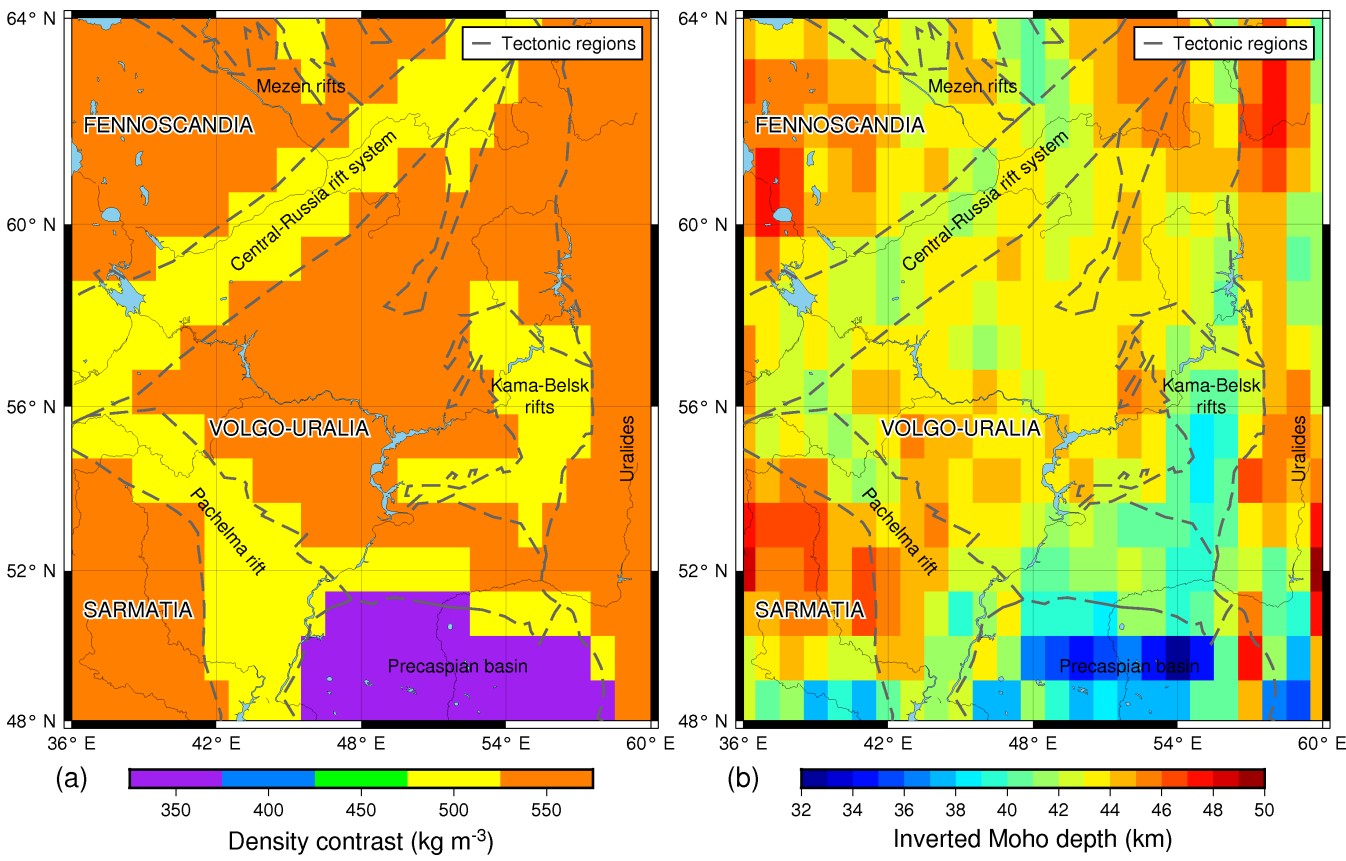

**Figure 5: (a) Density contrasts determined by using the algorithm of Haas et al. (2020) and (b) the Moho depth obtained through the gravity field inversion. Reference depth is equal to 45 km, crust-mantle density contrast of 550 kg m⁻³ is assigned to Archean cratons and Uralide orogen, 500 kg m⁻³ is assigned to Paleoproterozoic rifts, and 350 kg m⁻³ to Precaspian basin.**

### 4.2 Results of the forward modeling

The final product of the forward gravity modeling is the IGMAS+ 3D model of the Volgo-Uralian crustal structure. It includes the updated Moho model along with the main crustal interfaces. The constructed IGMAS+ model has a standard deviation of measured and calculated gravity equal to 8.0 mGal which corresponds to the correlation coefficient between the measured and calculated gravity of 0.91 (Fig. 6 a-c). For the vertical gravity gradient, the standard deviation is equal to 0.13 eotvos and the correlation coefficient is 0.81 (Fig. 7 a-c). This can be considered as an acceptable gravity fit for a regional crustal study (e.g. Sobh et al., 2019). Fig. 7 shows a long-wavelength residual in the gravity gradient field. This may potentially point out at the lateral density heterogeneity due to compositional change of the crust and lithospheric mantle of Volgo-Uralia which was not taken into account in the current model. The degree of such compositional density variations as well as possible causes of it is discussed in (Artemieva, 2003). The general look of the IGMAS+ 3D model with the locations of vertical sections is given in Fig. 8.

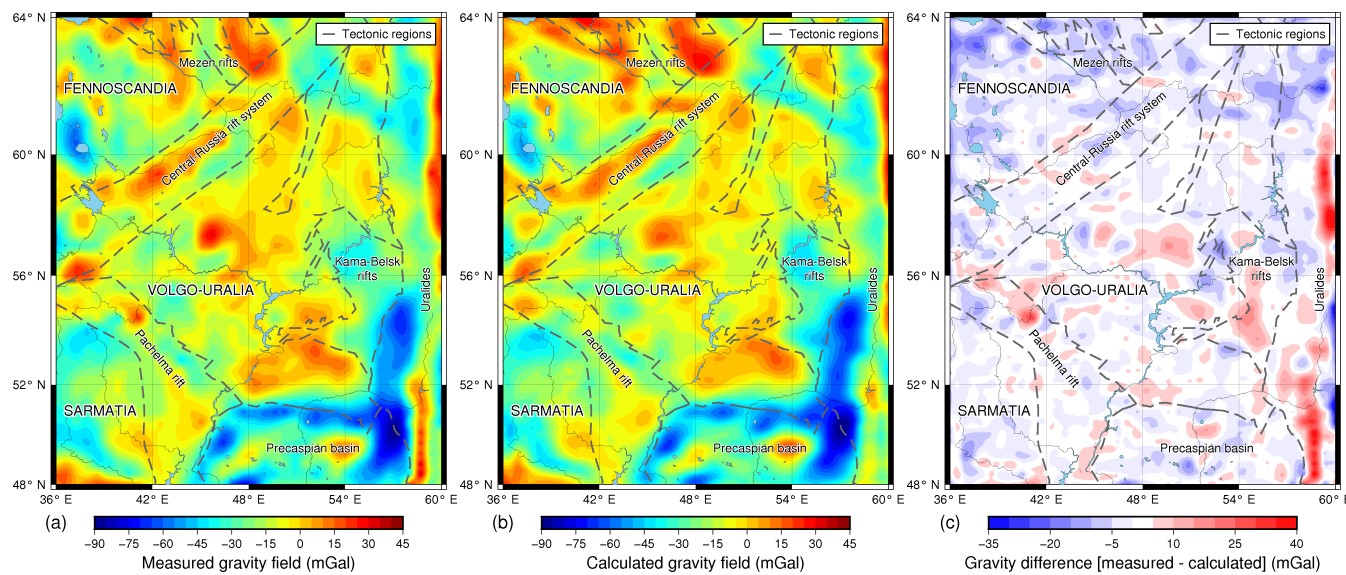

**Figure 6: Comparison between measured and calculated gravity fields. (a) XGM2019e Bouguer gravity anomaly (Zingerle et al., 2019). (b) Calculated Bouguer gravity anomaly from IGMAS+ 3D model. (c) The difference between measured and calculated gravity fields.**

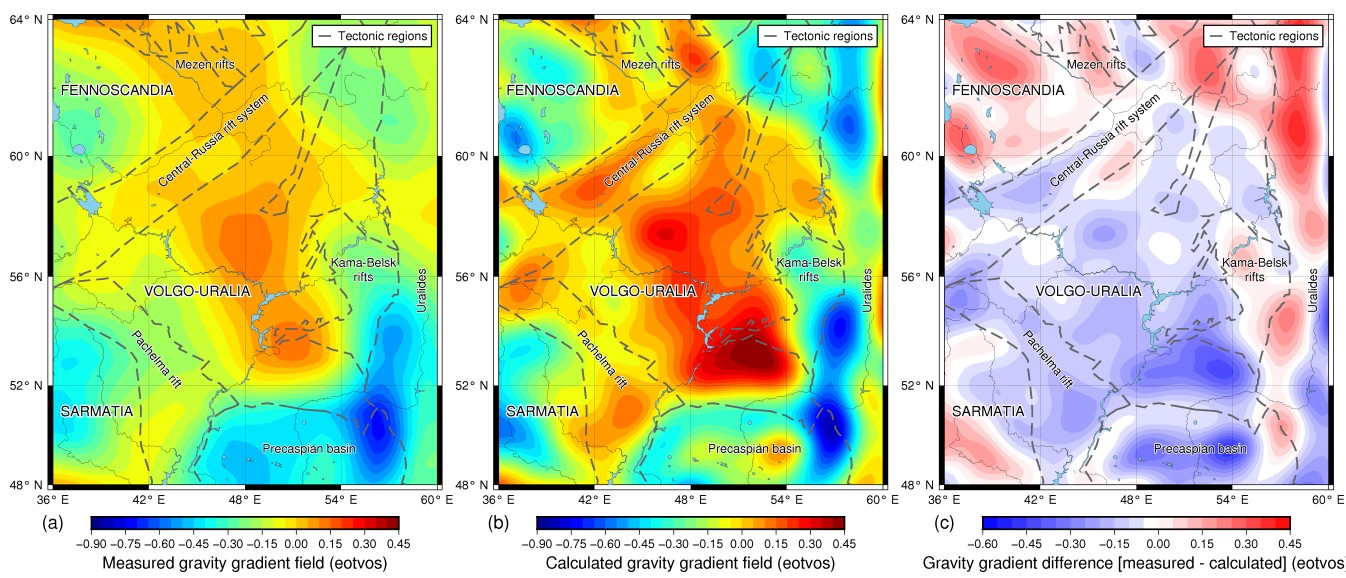

**Figure 7: Comparison between measured and calculated gravity gradient fields. (a) GOCE topographically corrected vertical gravity gradient at 225 km height (Bouman et al., 2016). (b) Calculated topographically corrected vertical gravity gradient field from IGMAS+ 3D model. (c) The difference between measured and calculated gravity gradient fields.**

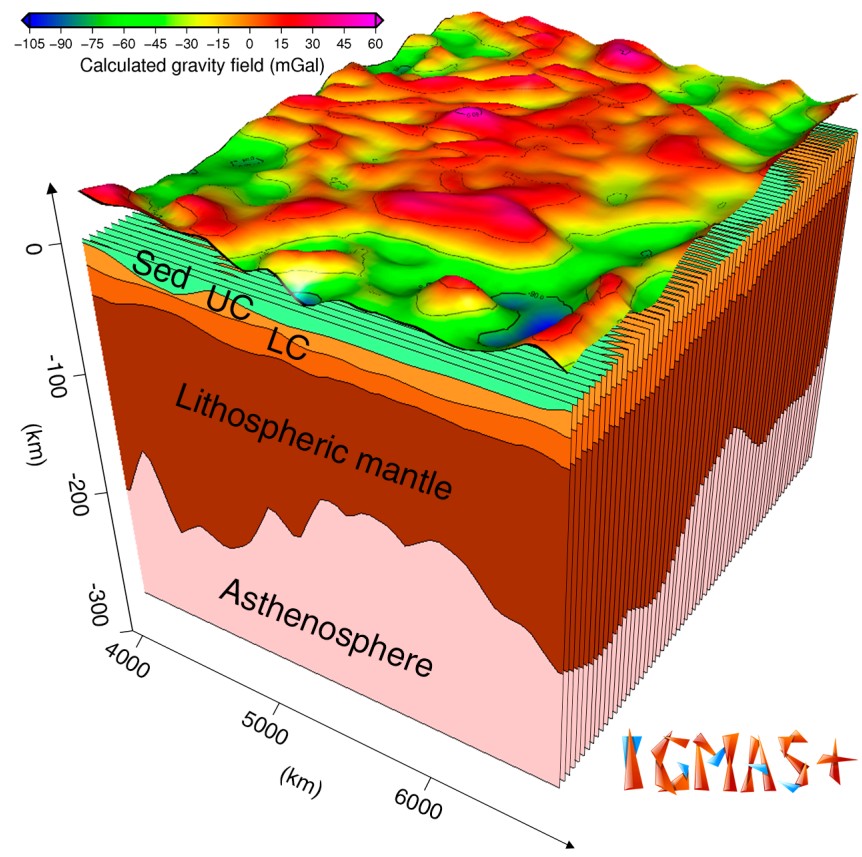

**Figure 8: A 3D lithospheric model of Volgo-Uralia developed in IGMAS+ software. It consists of 67 vertical sections which gives a spatial resolution of approximately 50 km. The model includes 5 main layers: sediments (Sed), upper crust (UC), lower crust (LC), lithospheric mantle, asthenosphere, and an additional 6th layer of underplating. The Bouguer gravity anomaly produced by the model is shown on top.**

Prior to reaching the aforementioned gravity fit, a considerable misfit of measured and calculated gravity revealed at the initial stage of forward modeling was interpreted and modeled as an underplated material (Section 3.3). This misfit has arisen after fitting the inverted Moho depth to the seismic data in the north-western portion of TATSEIS-2003 seismic profile. The depth difference between the seismic and inverted Moho depth is shown in Fig. 9a. Fig. 9b shows the difference between the Moho

 depth obtained in the forward modeling and the inverted Moho depth. From Fig. 9 one can see that Moho is much deeper in the center of the Volgo-Uralia in the area of interpreted underplating. This feature is not initially seen on the gravity inverted Moho depth map.

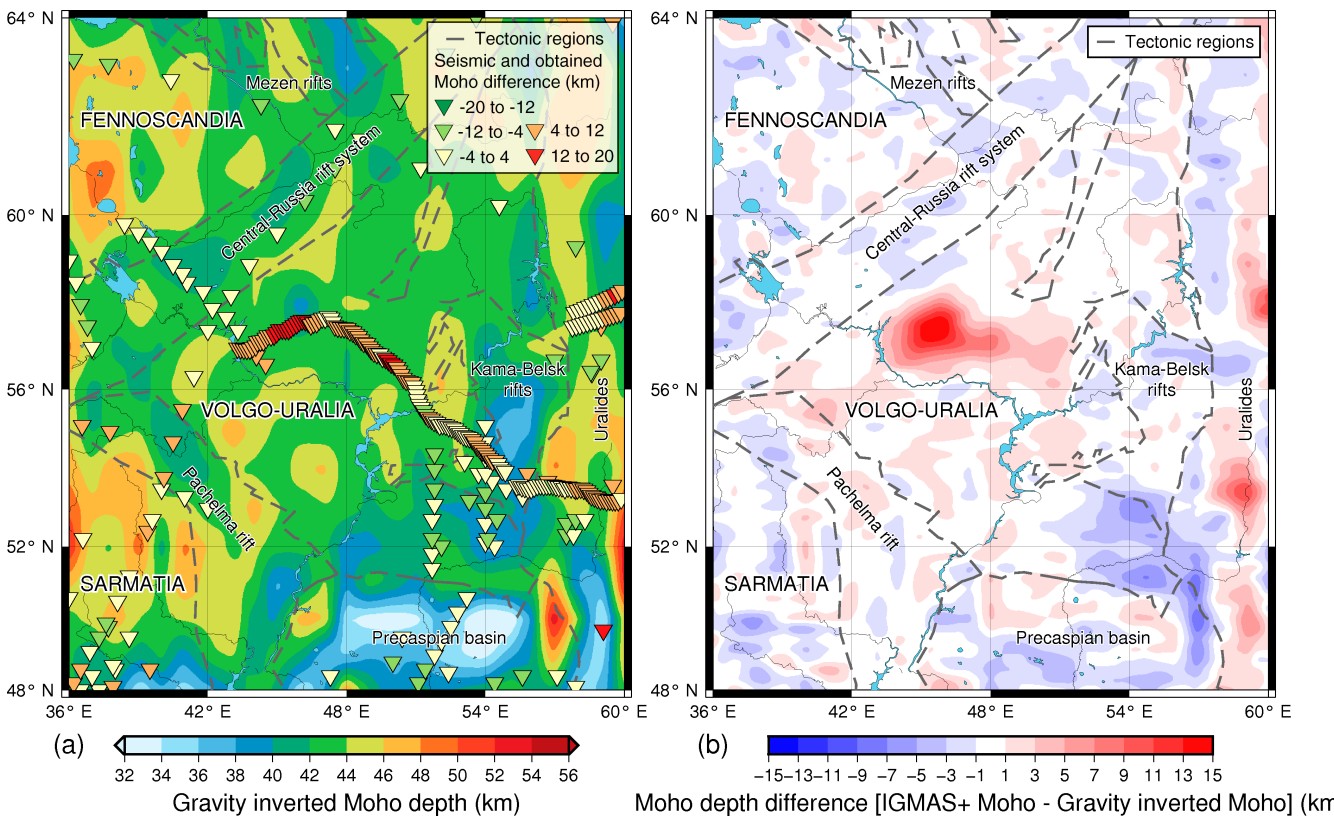

Figure 9: (a) Comparison of the inverted Moho depth and the seismic Moho estimates, where most of the difference is coming from the TATSEIS 2003 seismic profile in the center of the model. (b) Difference between IGMAS+ and gravity-inverted Moho depths.

The hypothesis of underplating in the area is not new. It was already suggested by Thybo and Artemieva (2013) and generally mentioned in the literature (Bogdanova et al., 2016, 2010; Mints et al., 2010). The recovered underplated body appears to be located on the north of the Tokmovo megablock under the Oka block (Fig. 1). This body is defined on a TATSEIS-2003 seismic profile as an acoustically transparent region (Trofimov, 2006). Mints et al. (2010) interpreted this feature as a domain of homogeneous mafic rocks partially metamorphosed into high-density granulites or eclogites at the basis of the so-called Vetluga synform. The isostatic calculations from Eq. (5) are also showing the high-density body with an average thickness of ca. 10 km which is clearly outlined by the area of isostatic imbalance in the center of Volgo-Uralia (Fig. 10). Other regions with the major mass deficits are located on the south-east of the map and are related to the Precaspian depression and South-Ural orogen. However, they do not correspond to any significant gravity misfit and are produced simply by the high deviation of the sedimentary and crustal thicknesses from the average values on the territory yielding higher values of mass imbalance.

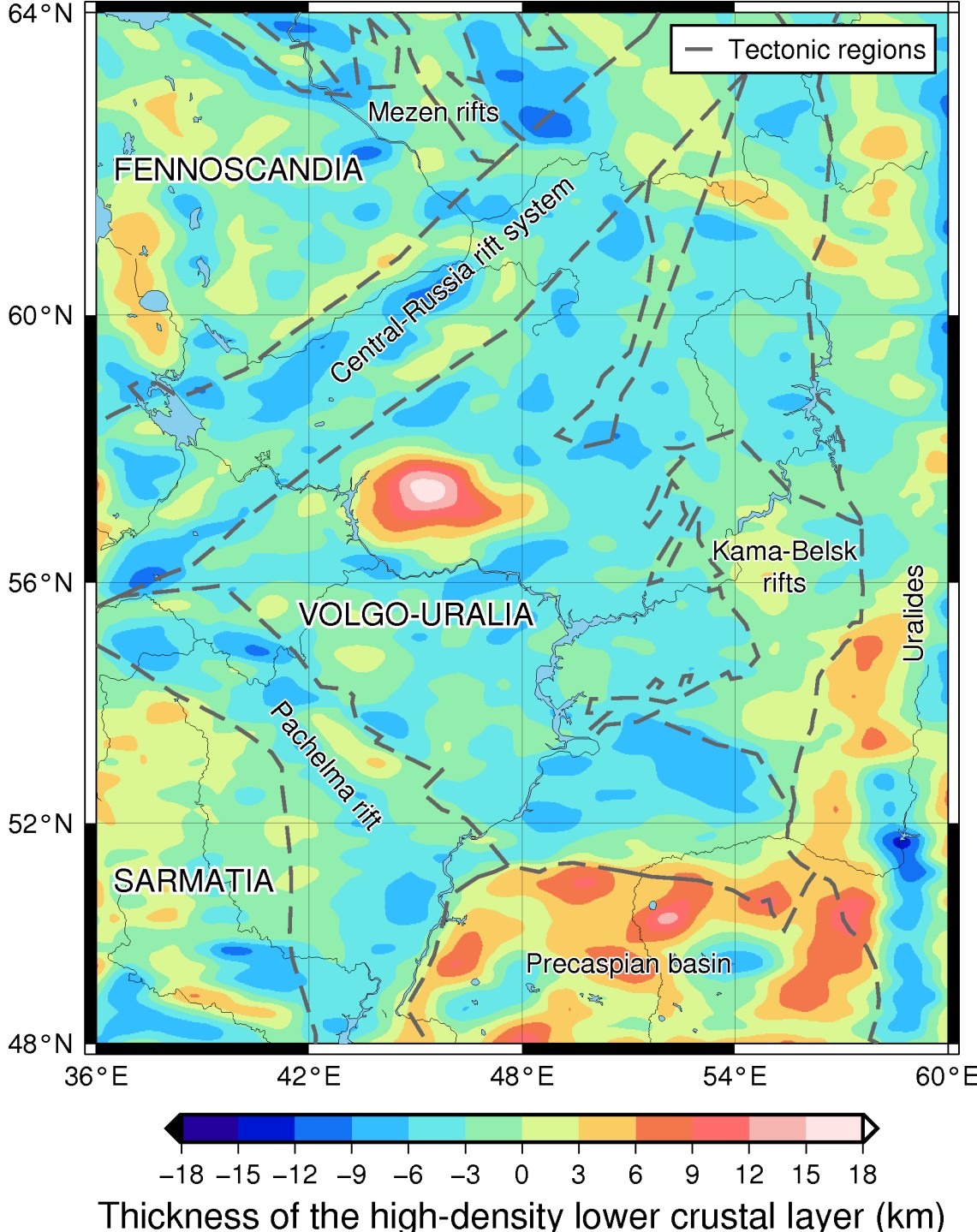

Thickness of the high-density lower crustal layer (km)

**Figure 10: Thickness of the high-density lower crustal layer from the isostatic calculations.**

The Moho depth of the developed IGMAS+ model shows a good agreement with seismic constraints (Fig. 11). The mean difference of seismic and modeled Moho is 0.75 km, the standard deviation is 3.31 km. This can be regarded as a satisfactory result as seismic Moho estimates usually are considered to have at least 2 km uncertainty (Ebbing et al., 2012). Therefore, at the end of the modeling, both seismic and gravity constraints were respected with the sufficient fit between measured and calculated gravity data.

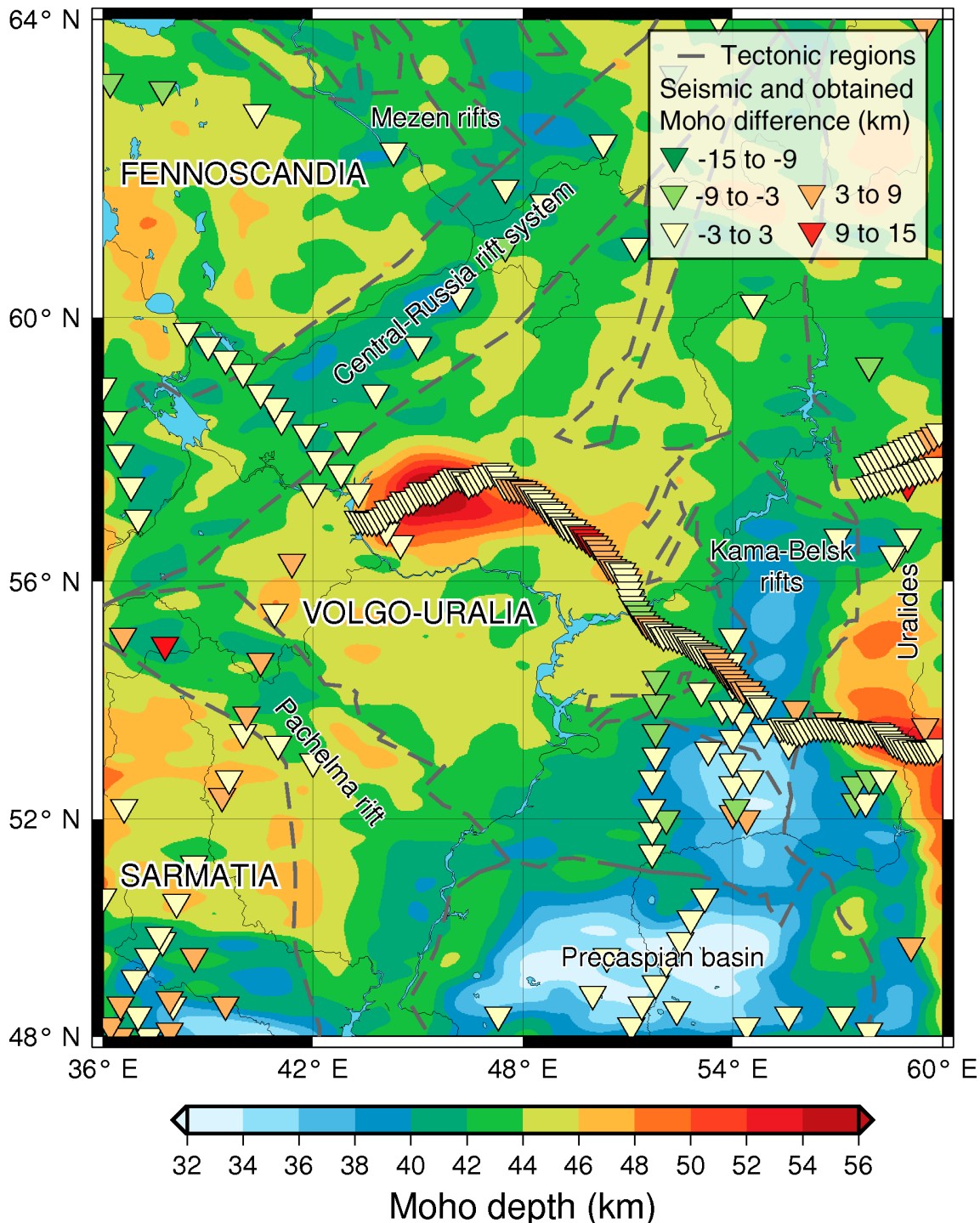

**Figure 11: Moho model of Volgo-Uralian subcraton obtained through the gravity inversion with laterally variable density contrasts (Haas et al., 2020) and subsequent forward gravity modeling with seismic and gravity constraints in IGMAS+ (Götze and Lahmeyer, 1988; Schmidt et al., 2020). The comparison between the model which was obtained in the process of gravity inversion and the IGMAS+ Moho model is shown in Figure S3 in the supplementary material.**

Most of the differences between seismic data and the Moho model developed in IGMAS+ are coming from the TATSEIS-2003 seismic profile. The seismic Moho depth along the TATSEIS-2003 and URSEIS-95 seismic profiles with respect to the model are shown in Fig. 12. As it is seen, within the TATSEIS profile seismic Moho has several steep troughs regarded as crustal roots (Artemieva and Thybo, 2013; Trofimov, 2006) which are not reflected in the satellite gravity field patterns. This case led us to a compromise solution: our Moho interface respects the main trends of Moho and at the same time smooths out

its sharp gradients providing a closer fit to the gravity constraints. It is visible that the Moho is deepest in the central cratonic region and it rises to the peripheral zone of Volgo-Uralia where it reaches the minimum depth of ca. 40 km under the thick sedimentary section of Sernovodsk-Abdulinsk Aulacogens. Then the crustal-mantle interface drops sharply below 50 km after reaching the Uralides.

A similar trend can be observed when looking at the north-south intersection of the eastern border of Volgo-Uralia (Fig. 13).

The crust is the thickest in the northern cratonic region which is away from the paleorifts. Whereas under the rift-like structures and in the cratonic area adjacent to them, the crust is thinning down. Particularly thin crust is observed in the south of the section under the Precaspian sedimentary basin.

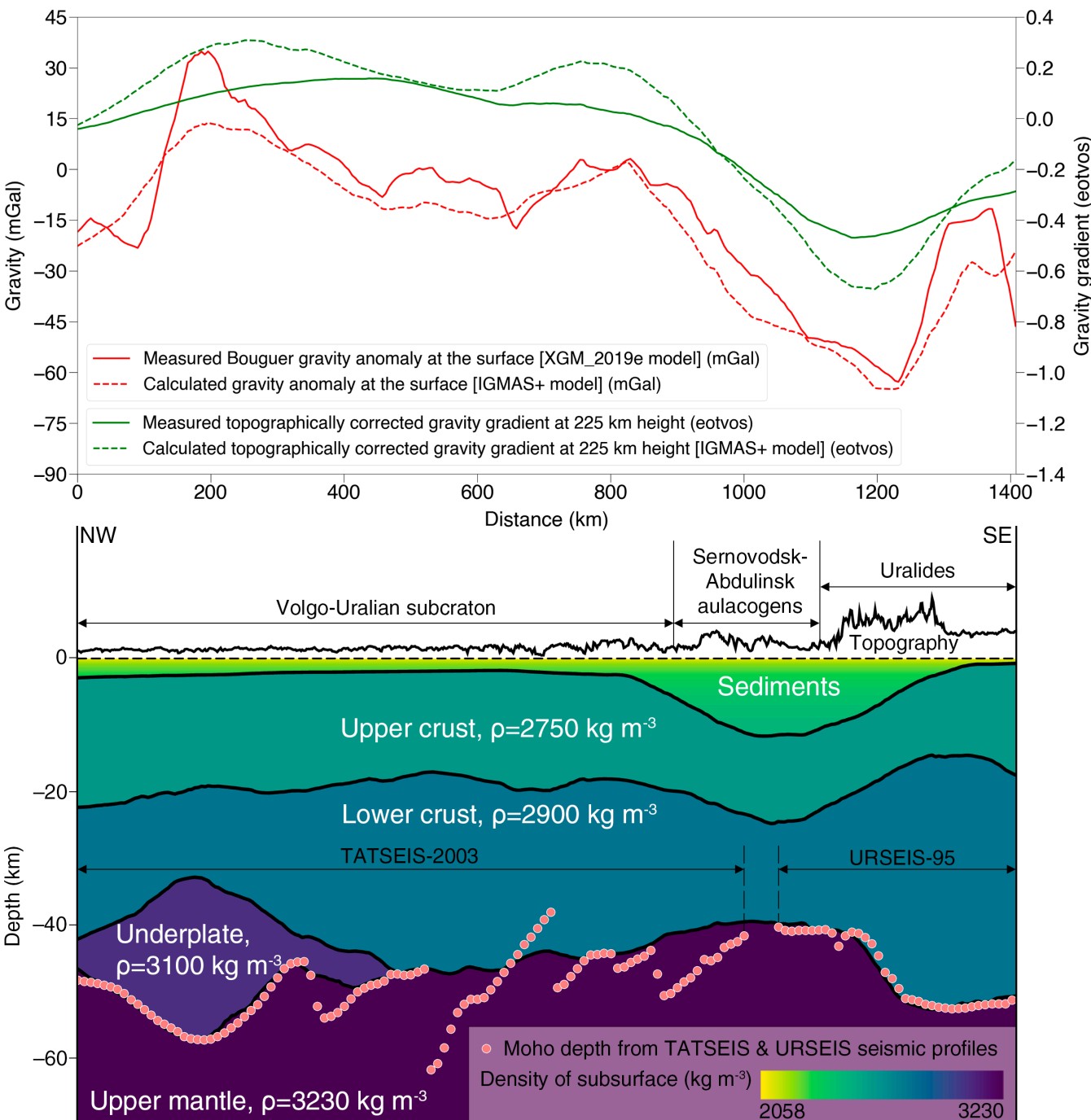

**Figure 12: Measured and calculated Bouguer gravity anomalies and the topographically corrected vertical gravity gradient anomalies from the crustal model (top) and IGMAS+ model cross-section along TATSEIS-2003 and URSEIS-95 deep reflection profiles (bottom) – see figure 3 for the reference on the map. Subsurface is vertically exaggerated by the factor of 10 and topography is by the factor of 100.**

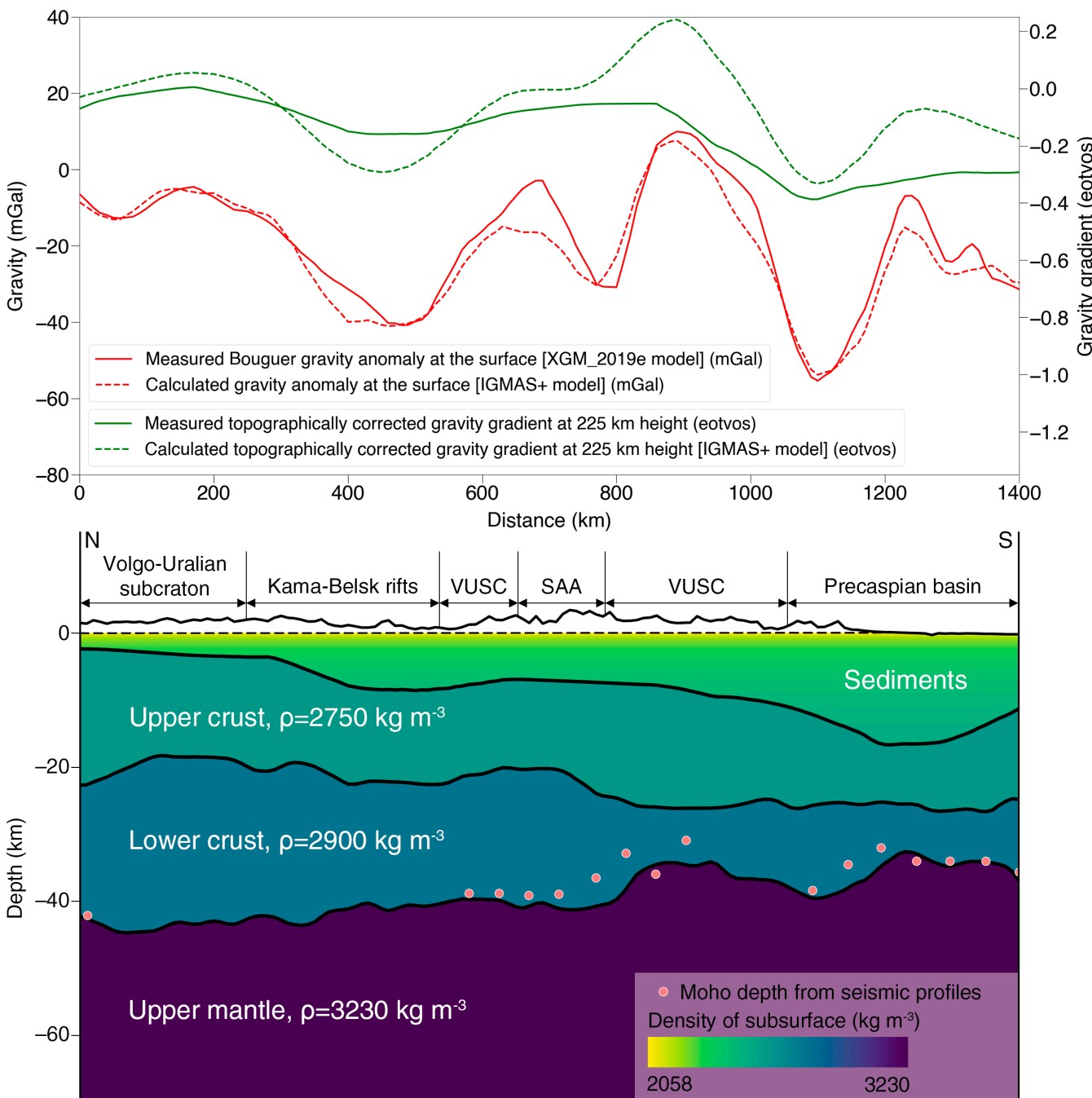

**Figure 13: Measured and calculated Bouguer gravity anomalies and the topographically corrected vertical gravity gradient anomalies from the crustal model (top) and IGMAS+ model north-south cross-section in the eastern part of Volgo-Uralia (bottom) – see figure 3 for the reference on the map. Subsurface is vertically exaggerated by the factor of 10 and topography is by the factor of 100. VUSC stands for Volgo-Uralian subcraton and SAA for Sernovodsk-Abdulinsk Aulacogens.**

The IGMAS+ model showed crustal thickness variation from 32 to more than 55 km in some areas. The thinnest crust with thickness below 40 km appeared on the Precaspian basin and Pre-Urals foredeep which correspond to the thickest sedimentary columns. A relatively thin crust was also found along the Central Russia rift system as well as in the north and south parts of the Pachelma rift. In the axial parts of both rifts, the thickness of the crust shrinks down to 40–42 km, whilst on the surrounding territory, the crust gains its thickness back up to 44–46 km. Thick crust is located underneath the Ural Mountains as well as in the center of the Volgo-Uralian subcraton. In each domain, crustal thickness exceeds 50 km. Overall, the developed model shows that within the EEC the Archean cratonic blocks are related to the thickening of the crust and Paleoproterozoic rifts are related to its thinning.

### 4.3 Comparison of the developed model to other regional Moho models

The resulting Moho model developed in IGMAS+ was cross-compared with the existing global and regional models which cover the studied region. For the comparison CRUST 1.0 global model (Laske et al., 2013), gravity-based GEMMA global model (Reguzzoni and Sampietro, 2015), and regional seismic EUNAseis model (Artemieva and Thybo, 2013) were selected. The difference between our model and the ones mentioned above is given in Fig. 14. It is clearly seen that the presented model is much deeper than GEMMA, and it has more similar depths to CRUST 1.0 and EUNAseis models. This is explained by the fact that our model as well as the CRUST 1.0 and especially seismic-only EUNAseis model are better constrained by the available seismic observations compared to the gravity-based GEMMA model.

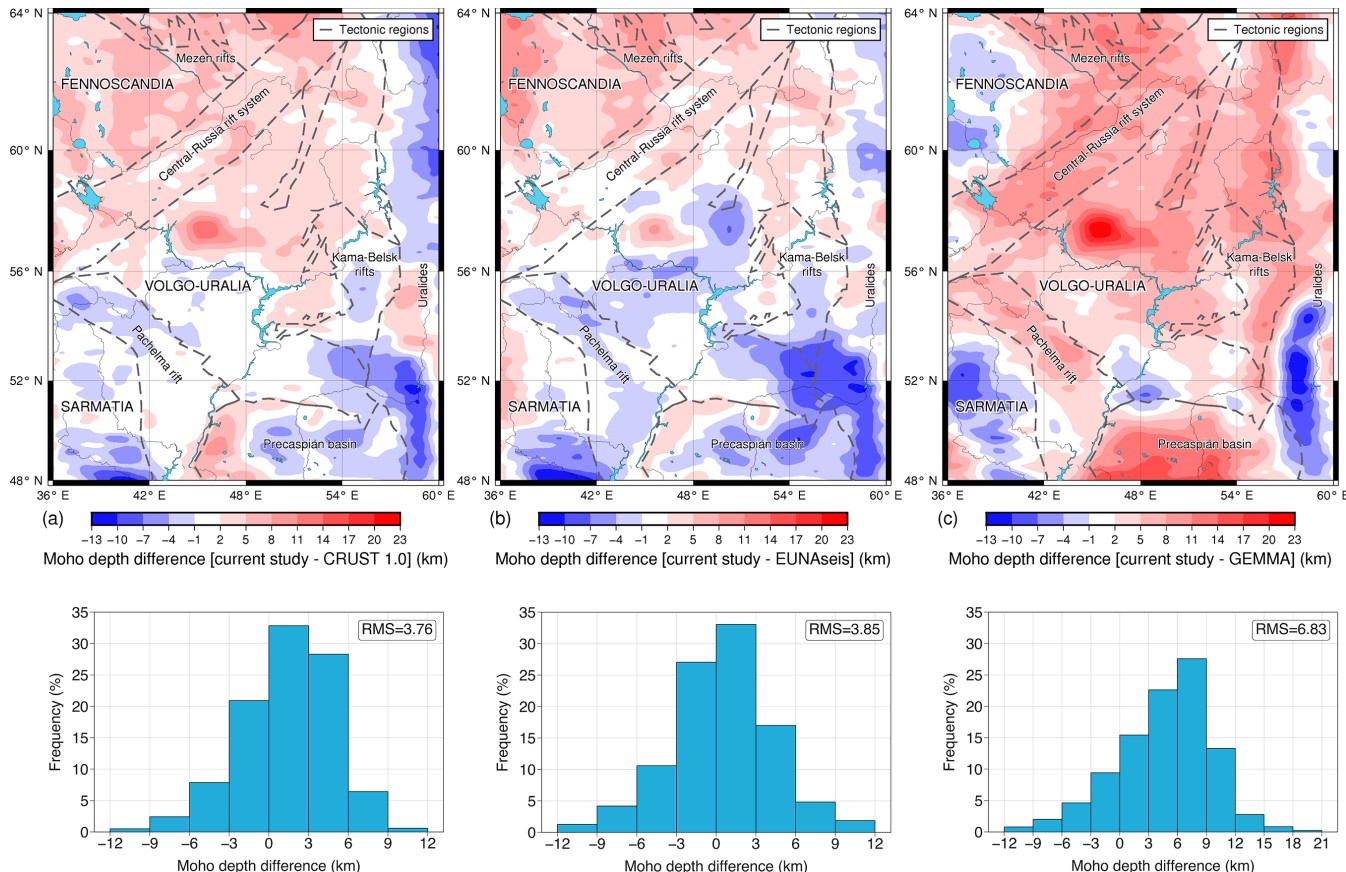

**Figure 14: Difference in Moho depths between (a) obtained model developed in IGMAS+ and CRUST 1.0 model by Laske et al.**
**(2013), (b) EUNAseis model by Artemieva and Thybo (2013), and (c) GEMMA model by Reguzzoni and Sampietro (2015). The top panel shows the maps of Moho depth residuals calculated as depth to Moho of the current study minus depth to Moho from the selected models in km. The bottom panel shows histograms of Moho depths differences in km.**

When comparing our model to EUNAseis and CRUST 1.0, it becomes obvious that the obtained model is relatively deeper on
the north-western part of the territory which corresponds to Fennoscandia. One of the possible explanations for this feature is that the south-western part of Fennoscandia has relatively sparse coverage with seismic stations. This could have led to the discrepancy of the Moho depth on this zone estimated by gravity and seismic-based methods. As a result, the model developed during this study and GEMMA gravity-based model show 5–10 km deeper Moho for south-western Fennoscandia compared to CRUST 1.0 and EUNAseis.

Another significant difference that is seen between our model and CRUST 1.0 is the thicker crust in the center of Volgo-Uralia in our model where the underplated body is recovered. Most probably, this difference has been revealed because the most recent seismic investigations on the Russian platform including the TATSEIS profile were not used in the compilation of

CRUST 1.0. One can see that the EUNAseis model which has an extensive seismic database for the Russian platform is closer to our model in the center of Volgo-Uralia where the underplated body is located.

The last conspicuous feature which is worth mentioning is the shallower Moho of the obtained model on the south-east of Volgo-Uralia as opposed to the EUNAseis model. Such anomaly arises because USGS seismic catalog and EUNAseis seismic database have been built independently and have certain differences in seismic Moho estimations in this region. Our model respects more the seismic estimates of Moho depth given by the USGS catalog on the south-east of Volgo-Uralia (Fig. 11) but diverges from EUNAseis Moho estimations showing 3–9 km shallower Moho in the south-eastern part of Volgo-Uralia and

south of Ural Mountains.

## 5 Conclusions

We presented a new crustal model of the Volgo-Uralian subcraton obtained through gravity inversion and thorough forward gravity modeling with seismic constraints.

The gravity inversion was performed using laterally variable crust-mantle density contrasts. Three different density contrasts

were estimated: 350 kg m$^{-3}$ for the Precaspian sedimentary basin, 500 kg m$^{-3}$ for Paleoproterozoic rifts, and 550 kg m$^{-3}$ for Archean cratons and Uralides. Reference Moho depth was equal to 45 km. As the result, we retrieved a gravity-inverted Moho depth of Volgo-Uralia. The gravity-inverted Moho model already exposed the major patterns of the crustal thickness variation in the area and was used as a preliminary layer in further 3D modeling.

Gravity field inversion was followed by 3D forward gravity modeling performed in IGMAS+ software. Here, additionally to

490 gravity-inverted Moho, sedimentary, crustal, upper mantle, and asthenospheric layers were included in the model. Seismic estimates of the Moho depth, as well as the Bouguer gravity anomalies from the XGM2019e gravity field model and topographically corrected GOCE gravity gradient served as the main constraints for the modeling. The 3D forward gravity modeling revealed a considerable gravity misfit in the central part of the study area. We interpreted this misfit as an underplated body which is supported by the isostatic calculations. This reinforces the hypotheses of an underplated body located on the top

of the Moho beneath the Oka block of Volgo-Uralia (Thybo and Artemieva, 2013).

The final crustal model respects all the main geological features of the Volgo-Uralian subcraton and its surroundings with Moho thickening in the cratons and under the Ural Mountains and thinning along the Paleoproterozoic rifts, Precaspian sedimentary basin, and Pre-Urals foredeep. The obtained crustal model will serve as a basis for further basin analysis and geothermal modeling.

## 500 6 Code and data availability

The code of Haas et al. (2020) for the gravity inversion with laterally variable density contrasts is available at https://github.com/peterH105/Gradient_Inversion.      The      data      used      in      this      study      is      available      at

https://doi.org/10.5281/zenodo.5701735 (Ognev et al., 2021). Figures and maps were plotted using ArcGIS Pro and Python with Matplotlib and PyGMT packages.

## 7 Author contributions

IO and JE designed the study. IO accumulated and processed the data, performed gravity field inversion, built the crustal model in IGMAS+, and visualized the results. JE supervised the work. PH wrote and shared the initial code for gravity inversion with laterally variable density contrasts. All the authors interpreted the results. All the authors contributed to the manuscript writing either by directly formulating text or giving feedback on figures or specific chapters.

## 8 Competing interests

The authors declare that they have no conflict of interest.

## 9 Acknowledgments

The presented work has been supported by the German academic exchange service (DAAD) and by the Ministry of Science and Higher Education of the Russian Federation under agreement No. 075-15-2020-931 within the framework of the development program for a world-class Research Center "Efficient development of the global liquid hydrocarbon reserves".

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
