# Peer review of "Crustal structure of the Volgo-Uralian subcraton revealed by inverse and forward gravity modeling"

_Solid Earth, 2021_

## Author Comment (AC1)

**Response to the comments of reviewer #1**

**Manuscript se-2021-98, Igor Ognev et al.**

**"Crustal structure of the Volgo-Uralian subcraton revealed by inverse and forward gravity modeling"**

Dear Reviewer,

We express a sincere appreciation of our manuscript's critical analysis and your valuable comments which allowed us to enhance the quality and the clarity of our manuscript. Please find below our responses to all your comments on a point-by-point basis. We use the blue color to distinguish our responses from your comments. The same blue color is used in the updated manuscript to show the corrected and added text.

**Major comments**

1. The "Tectonic setting" chapter should be extended by describing major evolutional steps of the large structural units. The Tatarian Arch must be mentioned since this structural element is characterized by the presence of large oil fields. Moreover, the Tatarian Arch is characterized by the presence of fluids within the crystalline rocks (e.g. Plotnikova, 2008; Plotnikova, I.N. New data on the present-day active fluid regime of fractured zones of the crystalline basement and sedimentary cover in the eastern part of Volga-Ural region. Int J Earth Sci (Geol Rundsch) 97, 1131–1142 (2008).) I would expect that these fluids should reduce the density of the upper crystalline crust in that region and possibly in other areas of the Volgo-Uralian subcraton. In this case, the influence of fluids should be discussed in the manuscript.

We extended the tectonic setting chapter with a description of major evolutional steps of the large sedimentary structural units with an emphasis on both North and South Tatar arches. We also added a figure in the supplementary material which is showing the location of the major Paleozoic tectonic structures of Volgo-Uralia.

The mentioned study of (Plotnikova, 2008) is indeed providing important insights on the fluids' circulation in the sedimentary cover and crystalline basement of the South Tatar arch and possible degassing of the basement. However, the measured density decrease of the oil deposits in the South Tatar arch is reported to be only ca. 2-5 kg m$^{-3}$ (fig. 12 in Plotnikova (2008)) which results in approximately 1-2 kg m$^{-3}$ of bulk rock density decrease within the oil-bearing part of the sedimentary section.

Such a small density change would not be reflected in the satellite gravity data or a large-scale global gravity model as XGM2019e. Even if we assume that such a density deficit can potentially be observed not only in the oil fields but also within several km of the crystalline basement section, it would still not be significant enough to be effectively represented in our lithospheric-scale model. Thus, even though the phenomenon of fluid circulation is important, it should be addressed in a specific local-scale study possibly with terrestrial gravity field measurements as constraints. At this point, it is out of the scope of the current study.

2. Another important point is related to the presence of the numerous huge salt structures in the Precaspian Basin due to mobilization of the Permian salt. Salt has a lower density than the sedimentary cover and, therefore, this feature of the Precaspian Basin must be discussed in the manuscript even if these salt structures are not completely covered by the model. I would expect that, at the regional scale, the influence of the low-density salt in the south should be still recognizable within the model area as well.

Agreed. We mention this feature of the Precaspian basin in section 3.2.2. And later-on distinguish the Precaspian as a separate tectonic unit with a unique density contrast.

3. A more detailed description of the IGMAS model extension out of the main model area must be given in the text. I mean - the lateral extension in order to minimize the edge effect. It is written that model has been extended by 2500 km. However, there is no information on how this extension has been done. Did the authors consider the main tectonic features for the extended parts, especially, towards the south where deep sedimentary basins are present beneath the Caspian Sea?

We added information about the model's lateral extension as compared to its dimensions and how the extension was performed in the section 3.3.

We did not model the tectonic structures outside the area of interest. In our case, the extension has been done solely to minimize edge effects. Practically it means that all the layers' thicknesses were extended by 2500 km from the area's of interest edges (see figure below).

[Figure]

Fig. 1. Lateral extension in IGMAS+.

4.      I have a question - Why the density contrast is also 550 kg/m3 beneath the Precaspian Basin? Even all old rift structures are characterized by the density contrast of 400 kg/m3, whereas the Devonian-Permian Precaspian Basin has the same density contrast as the Archean cratons.

Thank you for a reasonable question. Initially, we did not distinguish the Precaspian basin as a separate tectonic unit and practically treated it as a part of Volgo-Uralia. If we do distinguish it, we will have a slightly different density contrasts' lateral distribution with the 350 kg m$^{-3}$ density contrast in the Precaspian basin, and 500 kg m$^{-3}$ in paleorifts. This approach gives us the inverted Moho depth which is 1-5 km shallower in the Precaspian basin and 0.5-1 km deeper in the paleorifts as compared to the initial result.

Even though overall it gives a very similar Moho model for the whole study area, still the newly obtained model is closer to the seismic constraints in the area of the Precaspian basin and the finally obtained model in IGMAS+. We think, that it would be fair to distinguish Precaspian as a separate additional tectonic unit and keep this gravity inverted model.

5.      Another important point is the lower crustal body according to the isostatic calculations in Figure 9. This map reflects the Moho depth in Figure 10: the deep Moho is reflected by the thick lower crustal body and vice versa – the shallow Moho is reflected by this lower crustal body. I would like to admit that it is not a "body" in Figure 9. It is a high-density lower crustal layer which is characterized by the presence of several lower crustal bodies in places where this layer thickens. Therefore, "body" must be replaced by "layer" in Figure 9 and within the respective text.

Agreed, changed.

6.      The shape of the almost 17-km-thick solitary lower crustal body within the central part of the model area looks mysterious and must be discussed in more detail. There is a positive gravity anomaly over this body and the authors have mainly associated this anomaly with the lower crustal body. However, the shape of the anomaly (Fig. 6a) is more complex. I expect more discussion on the presence and the shape of this high-density lower crustal body. Is this body traced by the high-velocity body along the TATSEIS-2003 seismic profile? If there is no high-velocity body on the TATSEIS-2003 profile, the authors should explain why this body was not traced by the seismic data.

Yes, the body that we are defining has revealed itself on the TATSEIS-2003 seismic profile as a relatively acoustically transparent region. Mints et al. (2010) interpreted it as high-density metamorphic rocks of mafic composition at the base of the so-called Vetluga synform. We added this information into section 4.2.

The shape of the body in our case is dictated by the observed gravity misfit that was compensated by the body with the corresponding shape. Potentially it could stretch a little further as a thinner layer for a better gravity fit, but we decided to keep it realtively isometric as there are no further seismic indications of this body apart from the TATSEIS-2003 profile.

7.	Figure 11 is a very important figure, showing a difference between the results of gravity modelling and the seismic data along the TATSEIS-2003 and URSEIS profiles. However, I do not understand why the difference is so big beneath the thick sedimentary rocks. The gravity signal is the integral one and requires a differentiation at depth during the modelling. On the other hand, the seismic data are usually considered as a more reliable source since the seismic signal can be much easier localized at depth. I propose to use the seismic Moho configuration from the TATSEIS-2003 profile and add a lower crustal body beneath the thick sediments in order to fit the measured and the modelled gravity data rather than to model the Moho uplift in that area. Otherwise, the authors should explain why they do not trust the seismic data within this part of the TATSEIS-2003 on one hand. On the other hand, they almost precisely retrace the seismic Moho depth at the beginning of this seismic profile in their gravity model. The authors have written that "within the TATSEIS profile seismic Moho has several steep troughs regarded as crustal roots (Artemieva and Thybo, 2013; Trofimov, 2006) which are not reflected in the satellite gravity field patterns. This case led us to a compromise solution: our Moho interface respects the main trends of Moho…". From my point of view, the compromise solution should be to use a smoothed Moho depth, as the authors have done in between the deep Moho beneath the underplating and the modelled Moho uplift beneath the sediments. There are no indications for so strong Moho uplift according to seismic data as it has been modelled by the authors beneath the thick sediments. Of course, the velocity model along the TATSEIS-2003 can be theoretically not the best one in that area. But, in this case, the authors should explain why they think that seismic data are incorrect there.

We agree with the reviewer's concern about a certain discrepancy of seismic and modeled Moho beneath the thick sedimentary section and we appreciate the possible solution that the reviewer has proposed to this issue. Nevertheless, we must admit that such a discrepancy has arisen due to the kind of sedimentary model that we used.

During the review, it was found that our sedimentary model has approximately a 1-degree lateral North-East-to-South-West shift as compared to the EUNAseis original sedimentary model. Such a shift was a result of incorrect gridding at the initial stages of modeling. The updated model uses a correctly gridded EUNAseis sedimentary model. Eventually, we managed to keep the Moho depth as shown on the TATSEIS-2003 without having a significant gravity misfit in this area. Thus we did not need to add a high-density body in this region. See Fig. 12.

8.	Besides, the names of the tectonic units must be shown along the profile in Figure 11 in addition to names of the seismic profiles.

Agreed, added.

9.	I propose to show an additional cross-section through the 3D density model from the north to the south to see the transition from the internal cratonic areas towards the marginal Precaspian Basin.

Agreed, added as Fig.13.

10.	The boundaries of modelled area must be shown in Figure 1.

Added.

11.	Precaspian Basin is the more common English name of the "Pericaspian" Basin that has been used by the authors. I propose to use the "Precaspian Basin" rather than the "Pericaspian Basin".

Ok, thank you, we changed the "Pericaspian Basin" to the "Precaspian Basin" both in the figures and respective text.
* * *
We thank the reviewer for the thorough analysis of the manuscript and hope that we successfully addressed all the comments and questions.

With best regards on behalf of all the co-authors,

Igor Ognev.

**References**

Mints, M. V., Suleimanov, A. K., Babayants, P. S., Belousova, E. A., Blokh, Yu. I., Bogina, M. M., Bush, V. A., Dokukin, K. A., Zamozhnaya, N. G., Zlobin, V. L., Kaulina, T. V., Konilov, A. N., Mikhailov, V. O., Natapov, L. M., Piip, V. B., Stupak, V. M., Tikhotsky, S. A., Trusov, A. A., Filippova, I. B., and Shur, D. Yu.: Deep structure, evolution and minerals of the Early Precambrian basement of the East European Platform: Interpretation of materials on the reference profile 1-EU, profiles 4B and TATSEIS (in Russian), GEOKART: GEOS., Moscow, 2010.

Plotnikova, I. N.: New data on the present-day active fluid regime of fractured zones of crystalline basement and sedimentary cover in the eastern part of Volga-Ural region, Int J Earth Sci (Geol Rundsch), 97, 1131–1142, https://doi.org/10.1007/s00531-007-0274-z, 2008.

---

## Author Comment (AC2)

**Response to the comments of reviewer #2**

**Manuscript se-2021-98, Igor Ognev et al.**

**"Crustal structure of the Volgo-Uralian subcraton revealed by inverse and forward gravity modeling"**

Dear Reviewer,

We express a sincere appreciation for our manuscript's critical analysis. Your comments certainly allowed us both to enhance the quality of our manuscript and present the results of our work more clearly. Please find below our responses to all your comments on a point-by-point basis. We use the blue color to distinguish our responses from your comments. The same blue color is used in the updated manuscript to show the corrected and added text.

**Main questions and comments**

1. In the two step inversion procedure, it is sometimes difficult to follow which parameters are inverted from which data and where the seismic constraints are used. The schematic workflow of figure 2 could be completed by some additional details (Moho depth  $z_{ref}$ ,  $\Delta \rho$  range, a priori information, inverted parameters, hyperparameters...).

Thank you for your comment. Yes, we agree that it is better to complement the schematic workflow with more input information. We did it in the current version of the manuscript.

2. I. 145: "In addition, topographically corrected GOCE...": What about the data on satellite height? Aren't they topogrpahically corrected as well?

Yes, they are also topographically corrected. We made the writing clearer.

3. I. 156-157: For the LAB, could the highest density contrast boundary significantly differ from the isothermal boundary and to which extent could the depth difference impact the results?

The highest density contrast boundary could also be linked to the compositional boundary in case we have a significantly different composition between the lithospheric mantle and the asthenosphere. This could happen for example due to mafic depletion or on the opposite enrichment of the lithospheric mantle relative to the asthenosphere. To take this effect into account another approach for modeling is required which would allow the lateral density variation of the lithospheric mantle.

We would still expect the compositional boundary to be linked with the temperature, and thermal expansion phenomena, but it might be a lower temperature than what we used. According to our thermal calculations, a 100 °C degree temperature change would result in ca. 5 kg m-3 change of LAB density contrast. A much greater impact is caused by the

SPT density of the lithospheric mantle, which is ca. 10 kg m-3 of LAB density contrast change for the same change of SPT density. So, potentially if we have a lithospheric mantle relatively enriched with mafic components (e.g. with SPT density of 3370 kg m-3), our LAB density contrast can be greater than 30 kg m-3 which would impact our model in the area of such a compositional change.

In any case, this requires a different modeling approach with laterally variable lithospheric densities which was out of the scope of the current study.

4. I. 178: Could you add any reference for the value of the thermal expansion coefficient?

Yes, a reference is added to (Artemieva, 2007, 2019).

5. I. 179: "Slightly modifying Eq. (1)": Could you explain a little bit more how you go from Eq. (1) to Eq. (2)?

Yes, thank you for pointing this out. We added the explanation to the manuscript: "As the temperature at the Moho boundary does not contribute to the thermal expansion of the asthenosphere, we can slightly modify Eq. (1) to get in situ density of the asthenosphere by taking asthenosphere temperature as equal to LAB temperature".

Also, two errors were corrected in Eq. (1):

1) Eq.(1) had a typo: it should have a plus sign instead of a minus in the last member.

2) The initial calculations were performed with a different thermal expansion coefficient for the upper mantle and the asthenosphere. In the updated version of the manuscript, the same thermal expansion coefficient was used for these layers. This gives a LAB density contrast of 5 kg m-3 instead of initially used 10 kg m-3.

6. I. 220-222: Does one tectonic region correspond to a spatially constant  $\Delta \rho$ ?

Yes, it does. Now this point is specifically highlighted in the manuscript.

7. A few details on the inversion are missing and could helpful to the reader (eg. l. 230-231). What is the  $\Delta \rho$  step? On what are the  $\Delta \rho$  and depth ranges based? Could you provide any references? What is the grid spacing (see. point no 9)?

Yes, we added all this information in the manuscript at the end of the section 3.2.2.

8. I do not fully understand how the trade-off is made between the gravimetric inversion residuals and the fit to the seismic data (eg. I. 231, this is related to my point no 1). How is the Moho adjusted to the seismic data (eg. I. 253)? On which criteria? How do you deal with the uneven spatial distribution of the seismic data (eg. I. 277-278)? For this last question, I understand that you do not take it into account but that you are aware of this in the discussion of your results which seems fair to me.

1) The Moho was adjusted to fit the seismic data using two criteria:

A) When the Moho adjustment led to the enhancement of the gravity fit to both components of the gravitational potential (gravity and gravity gradient) or one of the components without significantly losing the fit to the other.

B) when the seismic data was showing consistently different Moho depths compared to the inverted Moho like on one of the digitized seismic profiles (see TATSEIS-2003) which had to be respected.

2) In our updated model we do take uneven data coverage into account by decreasing the resolution of the traced seismic profiles (TATSEIS-2003, URSEIS-95, ESRU, UWARS) from 10 to 40 km to make it comparable to the one of the USGS seismic catalog. In line with the addition of the complementary 4th tectonic unit (see Precaspian Basin at the end of section 3.2.2) in our gravity inversion, it led to a slightly different result in terms of the density contrast lateral distribution (Fig. 5). But overall the newly obtained inverted Moho model remains very close to the initial one.

9. I. 238 "extended by 2500 km": This value is to be compared to the extent of the study area and the size of the elements of the model.

**Added.**

10. I. 240-241: "triangulated polyhedrons in-between vertical cross-sections": I am not familiar with kind of "hybrid" modelling. Out of curiosity, what are the advantages?

The main advantage is a fast feedback loop. A user can go on each of the sections, modify the positions of the polyhedrons' vertexes and immediately see the change of gravity fit of measured and calculated anomalies on this section or the entire map. Another advantage of the software is its capability of displaying multiple additional data on your sections like seismic constraints on depth to Moho in the form of points or interpreted seismic sections as georeferenced images. You can then easily adjust your interfaces' geometry to better fit all the constraints.

11. Fig. 6-7: It seems that there is a long wavelength signal left in the residuals. Has a regional/long wavelength signal been removed from the data? If not, why not? I think that it could help but the relevence of such correction might be questionable at this scale. What do you think about this long wavelength signal and the (ir)relevance of a correction of the data before the inversion?

Such a long-wavelength signal in the residuals may be a result of a lateral density heterogeneity of the East European Platform (Artemieva, 2003). This lateral density heterogeneity of the EEP can be is linked to its compositional change. So it is probably a good idea to work with the data as it is, because introducing a correction for not-fully known compositional variation of the crust can be rather problematic.

12. I. 321: "also manifested a considerable misfit": I am confused with this statement while the fit was described as "acceptable" a few lines before.

We agree it may seem confusing. This sentence was rephrased to be clearer.

13. The discussion and conclusion mention that "the 3D forward gravity modeling revealed a considerable gravity misfit in the central part of the study area" (I. 408-409). However, I could not see this feature on the figures shown... until I saw the supplementaries. In think that figure S2 deserves to be moved to the main article as the maps show features that are widely discussed. You might combine fig. 10 and S2b if it is not too small to remain readable.

We modified the S2a by adding differences of Moho depth between the inverted and seismically obtained Moho, combined it with S2c, and added this figure in the manuscript as Fig. 9 along with respective discussion in section 4.2.

14. Fig. 9: I do not understand this figure with negative thicknesses. Are these variations?

To obtain this figure we performed isostatic calculations of mass imbalance from Eq. (5) and assumed that the areas of negative imbalance correspond to the high-density material in the lower crust. Then we found the thickness of this material by dividing the obtained mass imbalance by the density difference between the regular lower crust and the assumed high-density portion of it. It was 2900-3100=-200 kg m-3. It was done to estimate the possible thickness of the underplated body assuming the studied region is isostatically compensated.

15. Fig. 10: The map would be more readable if the same information and colorscale was shown as background and in the triangles. If you showed the seismic Moho depth in the triangles, the reader would more easily see the areas where the color of the triangle significantly differs from the background color.

We tried this way of presenting the result as well (see figure below). In this way even when the obtained model differs from the seismic constraints, the colors of triangles and the map still remain very close on a color pallet. It makes it hard to notice the places of difference. We would prefer to keep our initial way of presenting the differences between seismic data and the obtained model as it seems that in this case, the differences are a little more conspicuous.

---

## Author Response (AR2)

**Author's response**

**Manuscript se-2021-98, Igor Ognev et al.**

**"Crustal structure of the Volgo-Uralian subcraton revealed by inverse and forward gravity modeling"**

Dear Topical Editor,

Below you will find a point-by-point response to the two remaining comments. Our responses are distinguished from comments by the blue color. We attach the updated manuscript where we also corrected a few minor mistakes in the references, text, and figures.

Thank you for your time and patience,

With best regards,

Authors.
* * *
**Remaining comments**

1.     The Moho temperature is taken to be constant at 500 ºC. Please discuss that Moho temperature in Earth is not constant at different depths and that this assumption is therefore a simplification.

Agreed. We added two sentences regarding that and cite (Mareschal and Jaupart, 2013) on this issue.

2.     Please define DeltaLoad and g in equation 5.

Added.
* * *
We thank the reviewers for the thorough analysis of the manuscript. Their analysis and comments helped greatly improve the quality of the paper. We also thank the editor for carefully monitoring the review process.

With best regards on behalf of all the co-authors,

Igor Ognev.

**References**

Mareschal, J.-C. and Jaupart, C.: Radiogenic heat production, thermal regime and evolution of continental crust, Tectonophysics, 609, 524–534, https://doi.org/10.1016/j.tecto.2012.12.001, 2013.